

# A Review and Synthesis of Future Earth System Change in the Interior of Western Canada: Part I - Climate and Meteorology

Ronald. E. Stewart[1], Kit K. Szeto[2], Barrie R. Bonsal[3], John M. Hanesiak[1], Bohdan Kochtubajda[4], Yanping Li[5], Julie M. Thériault[6], Chris M. DeBeer[7], Benita Y. Tam[2], Zhenhua Li[5], Zhuo Liu[1], Jennifer A. Bruneau[1], Sébastien Marinier[6] and Dominic Matte[8]

[1]Department of Environment and Geography, University of Manitoba, Winnipeg, Manitoba, Canada (ronald.stewart@umanitoba.ca, john.hanesiak@umanitoba.ca, zhuo.liu@umanitoba.ca; umbrun27@myumanitoba.ca)
[2]Climate Research Division, Environment and Climate Change Canada, Toronto, Ontario, Canada (kit.szeto@gmail.com - retired, benita.tam@canada.ca)
[3]Watershed Hydrology and Ecology Research Division, Environment and Climate Change Canada, Saskatoon, Saskatchewan, Canada (barrie.bonsal@canada.ca)
[4]Meteorological Service of Canada, Environment and Climate Change Canada, Edmonton, Alberta, Canada (bob.kochtubajda@canada.ca)
[5]Global Institute for Water Security, University of Saskatchewan, Saskatoon, Saskatchewan, Canada (yanping.li@usask.ca, zhenhua.li@usask.ca)
[6]Centre ESCER, Department of Earth and Atmospheric Sciences, Université du Québec à Montréal, Montréal, Quebec, Canada (theriault.julie@uqam.ca, marinier.sebastien@courrier.uqam.ca)
[7]Centre for Hydrology and Global Institute for Water Security, University of Saskatchewan, Saskatoon, Saskatchewan, Canada (chris.debeer@usask.ca)
[8]Niels Bohr Institute, University of Copenhagen, Copenhagen, Denmark (dominic.matte@nbi.ku.dk)

*Correspondence to*: Ronald E. Stewart (ronald.stewart@umanitoba.ca)

**Abstract.** The Interior of Western Canada, up to and including the Arctic, has experienced rapid change in its climate, hydrology, cryosphere and ecosystems and this is expected to continue. Although there is general consensus that warming will occur in the future, many critical issues remain. In this first of two articles, attention is placed on atmospheric-related issues that range from large scales down to individual precipitation events. Each of these is considered in terms of expected change organized by season and utilizing climate scenario information as well as thermodynamically-driven future climatic forcing simulations. Large scale atmospheric circulations affecting this region are generally projected to become stronger in each season and, coupled with warming temperatures, lead to enhancements of numerous water-related and temperature-related extremes. These include winter snowstorms, freezing rain, drought as well as atmospheric forcing of spring floods although not necessarily summer convection. Collective insights of these atmospheric findings are summarized in a consistent, connected physical framework.

## 1 Motivation and objective

Climate and its changes are having huge impacts everywhere. A particular 'hotspot' in Canada in terms of recent temperature changes and projections of continuation is the central part of western Canada and its extension to the Arctic Ocean (DeBeer et




al., 2016) Although there is widespread consensus that warming will continue, there is considerable uncertainty in its magnitude and distribution in time and space. There is even greater uncertainty in terms of precipitation although it is very likely that there will be less snow and more rain, and the north will become wetter as compared to the south.

All of these changes have a huge impact on water resources, cryosphere and ecosystems. In terms of hydrology, this includes the amount of water as well as the timing of its peak flow; in terms of the cryosphere, this includes the fate of numerous glaciers, regions of permafrost and the duration and amount of snow; in terms of ecosystems, this includes movement of grasslands, tundra, shrubs and boreal forests.

These issues were a critical motivation for the development of the Changing Cold Regions Network (CCRN). This 5-year (2013-18) research program aimed to understand, diagnose and predict interactions amongst the cryospheric, ecological, hydrological and climatic components of the changing Earth system at multiple scales with a geographical focus on Western Canada's rapidly changing cold interior (DeBeer et al., 2015). Its area of concern is shown in Fig. 1, and this includes the Saskatchewan and Mackenzie River systems; all geographic locations and terms referred to in this article are also indicated.
CCRN represents a regional hydroclimate project that was formed under the auspices of the Global Energy and Water Exchanges (GEWEX) project of the World Climate Research Programme.

Initially, CCRN collated many studies documenting a wide variety of variables to best characterize recent change over this region (DeBeer et al., 2016). Widespread change was documented in air temperature, precipitation, seasonal snow cover,
mountain glaciers, permafrost, freshwater ice cover, and river discharge. Increases in air temperature were the most notable within the domain, with annual values rising on average 2°C throughout the western interior since 1950. This increase has been associated with changes to precipitation regimes and unambiguous declines in snow cover depth, persistence, and spatial extent. Consequences of increasing air temperatures have caused mountain glaciers to recede at all latitudes, permafrost to thaw at its southern limit, and active layers over permafrost to thicken. Despite these changes, integrated effects on annual
streamflow amounts are complex and often offsetting, but the timing of the spring freshet has, in general, advanced to earlier in the year as a result of rising air temperatures and earlier snowmelt.

As indicated above, one of the key goals of CCRN is linked with future conditions. Many articles have utilized climate model projections using, for example, Coupled Model Intercomparison Project Phase 5 (CMIP5) information (Taylor et al., 2012)
and Intergovernmental Panel on Climate Change (IPCC) reports (such as IPCC, 2013). However, more insight is required into the processes and drivers of this change over the CCRN region of interest including, for example, physically-based examination of features in the future and conceptual models of change. Such insight provides guidance as to the reliability of models and to research focal points for improving future projections.



We follow this avenue by examining projected changes in several, often related, phenomena in a physically-consistent manner through a cascade of scales and through physical understanding. This article is a natural follow-on to the summary of recent change over the CCRN region (DeBeer et al., 2016) but also to earlier synthesis articles over the Mackenzie River basin studies (Szeto et al., 2007; Woo et al., 2008) and over the Canadian Prairies during its 1999-2005 drought (Hanesiak et al., 2011).

With this background, our objective is to summarize and synthesize our collective assessments of future conditions across the CCRN domain. The breadth of CCRN is so large that this overall issue cannot be addressed within one article. It is broken into parts as follows:

Part 1: climate and meteorology

Part 2: terrestrial ecosystems, cryosphere, and hydrology

The specific objective of this first article is to illustrate how changing large scale conditions will affect regional and storm scales with a general, although not exclusive, focus on precipitation-related phenomena and its extremes. This approach facilitates increasing our insights into regional hydroclimate response to projected large-scale circulation changes. Overall

warming will be associated with changes in large scale atmospheric circulations and moisture, but it is critical to quantify these changes and to examine consequences on smaller scale features. The article is comprised of key outcomes from completed studies, new analyses as well as an overall synthesis.

The article is organized as follows. Section 2 provides a summary of model datasets and analysis, Sect. 3 examines issues at

seasonal scales, Sect. 4 addresses phenomena in more detail within the cold season, spring and early summer, as well as summer periods. Section 5 presents a synthesis and Sect. 6 contains the concluding remarks and sets the stage for the second article focused on surface-related issues.

## 2 Model datasets and analysis

Given that the main objective of this article is to attain a deeper and more coherent understanding of different regional aspects

of climate change in western Canada, we adopt the notion that climate change alters large-scale circulations that govern much of the climate variability at regional scales. This premise provides a perspective for analyzing the CMIP5 data. Monthly projections using the RCP8.5 scenario from 39 CMIP5 models were analyzed to gain insight into the cascading processes that link regional responses to changes in the large-scale circulations. In particular, this information was used to generate ensemble mean, median as well as the top and bottom 25$^{th}$ percentile values. The evolution of several standard and derived variables

over the 21$^{st}$ century were evaluated and there was a particular focus on differences between mean 2081-2099 and mean 1981-2000 values.



Given the distinct seasonal differences in the study region's present and projected climate, results are organized largely by seasonal change. In addition, it is well-known that the climate of the region is strongly influenced by teleconnection patterns, particularly the Pacific North American (PNA) pattern (Wallace and Gutzler, 1981) during the cold season (see for example, Table 2 of Szeto, 2008), and quasi-stationary upper air circulation features over the northwestern U.S. during the warm season

5 (Shabbar et al., 2011; Szeto et al., 2016). Emphases are thus placed on the analysis of future changes of such large-scale circulation features. In addition to focusing on the most prominent large-scale circulations that affect the region, this approach also simplifies the interpretation of the influences of these features on regional warming distribution and future high impact climate extreme events in western Canada. As appropriate, this insight is supplemented by that from previous, related studies as well as analyses conducted with CMIP5 daily data.

Global climate model (GCM) information is essential but additional datasets are needed because the analysis considers regional and storm scales. New analyses were conducted by using regional data to fill in critical research gaps that had not been addressed in previous regional studies. Those include traditional dynamically downscaled regional and storm scale datasets such as CRCM5 (Canadian Regional Climate Model version 5, Martynov et al., 2013 and Šeparović et al., 2013), NARCCAP

15 (North American Regional Climate Change Assessment Program, Mearns et al., 2013), NCEP/NCAR (National Centers for Environmental Prediction/National Center for Atmospheric Research) re-analysis (Kalnay et al., 1996), and Environment and Climate Change Canada (ECCC) weather station information.

In addition, the pseudo-global warming (PGW) approach described by Liu et al. (2016C) is used to examine the impact of

20 thermodynamic forcing on recent patterns and phenomena. Simulations were carried out with the WRF (Weather Research and Forecasting) model version 3.6.1 at 4 km spatial resolution for the period 2000-2013 (control, CTRL) and then repeated under an assumption of PGW (2071-2100). The projections utilized a multi-model ensemble-mean change signal under the RCP8.5 emission- scenario and covered a substantial portion of North America but only up to approximately 55-57°N; this dataset is referred to as WRF HRCONUS (High-Resolution Contiguous United States) as described by Liu et al., (2016C).

As appropriate, the historical period is generally considered to be 1981-2000 although some Fourth Assessment Report (AR4) analyses have used 30 year averages with 1971-2000 as the base period. As noted above, the historical period used with WRF is limited to 2000-2013. Temporal changes are examined over various domains although one focus is the southern Prairies bounded here by the latitude-longitude box (95-115°W, 47.5-55°N).



## 3 Large and regional scale patterns

### 3.1 Autumn

Projected mid-tropospheric (500 hPa) circulation changes resemble a westward shifted negative PNA pattern (Fig. 2a). In particular, an anomalous trough is projected to occur above British Columbia/Yukon and extending into the Pacific off the

west coast of the U.S. while an anomalous high is projected to occur over the southwest vicinity of the Aleutian Islands. These circulation anomalies, as well as those to be discussed below for other seasons, are deep structures that extend to the top of the troposphere. Similar circulation patterns are typically found during negative PNA conditions with an anomalous low centered above the southern Prairies, and a high above the Aleutians. Corresponding changes in anomalous mean sea-level pressure (MSLP) include a projected high centered just east of the upper high and a trough that extends from the Arctic Ocean into

central north Canada. Circulation changes at higher levels effectively reduce the waviness of the upper flow and jet stream, and thus reduce the potential for synoptic scale disturbances that might affect western Canada. Although inter-model spread as measured by the interquartile range is large, a decreasing trend of SON PNA index, especially at around mid-century, is exhibited in most model projections (Fig. 3a).

Projected regional climate responses to the circulation changes are consistent with those found during negative PNA, but shifted in association with the projected circulation features. For example, enhanced cold air advection into northwestern Canada is induced by the anomalous flow associated with the surface North Pacific high. However, due to the westward-shifted location of the high, the cooling is limited to northern regions when compared to typical negative PNA conditions. The surface high also enhances low-level flows towards the Pacific coast of Canada, which when combined with the upper low

over British Columbia, would substantially enhance precipitation over the coastal regions (Fig. 2d). The enhanced cross-barrier flow and associated precipitation induces subsidence and adiabatic warming over the Prairies (Szeto, 2008). The warming over the south and cooling over the north effectively reduces the S-N gradient of net anthropogenic warming (Fig. 2c) and offsets its negative impacts on the low-level background baroclinicity and synoptic storms in southwestern Canada.

Quasi-geostrophic theory (e.g., Holton, 1979) predicts that cyclone activities would be enhanced in the downstream vicinity of the upper anomalous trough, i.e., over southwestern Canada. Despite the potential increases in autumn cyclones and atmospheric moisture in the warming environment, Prairie precipitation is projected to increase by only ~10%. This is likely related to the significant depletion of Pacific moisture over the coastal mountains and enhanced lee-side subsidence that are associated with the enhanced cross-barrier flow discussed earlier. As a result, although the precipitation increase is statistically

significant (i.e., the increase is larger than the natural variability of historical precipitation for the region), it is substantially lower than the rate predicted by Clausius-Clapeyron scaling (~24%, based on temperature increases over the North Pacific and Atlantic Oceans, the main moisture source region for autumn Prairie precipitation). These results indicate that the complex



topography of the region could play an important role in affecting the autumn precipitation over the Prairies under this warming scenario.

## 3.2 Winter

The winter anomaly pattern is projected to be characterized by a pronounced upper low centered in the eastern Pacific and

enhanced ridging over central northern Canada (Fig. 4a). In contrast to projected SON changes, the circulation change resembles an eastward-shifted positive PNA pattern.  In fact, an increasing trend in the DJF PNA index, particularly during the mid-century,  is exhibited in most model projections (Fig. 3b). These results might be related to those of Zhou et al. (2014) who showed that the eastward shift of tropical convective anomalies under climate warming would cause the ENSO-forced winter PNA pattern to move eastward and intensify. These projected changes effectively enhance the waviness of the upper

flow and jet over the North Pacific by strengthening and broadening the North Pacific upper trough towards the west coast of the U.S. while broadening the climatological upper ridge and making the flow more zonal over northwestern Canada. Some changes at the lower levels (Fig. 4b) are similar to those found in positive PNA conditions with an anomalous trough extending from the Aleutians into areas off the west coast of North America. But, a strong anomalous surface ridge that is typically located over the western U.S. under positive PNA conditions is projected to be centered over southwestern Canada.

Similar to positive PNA conditions, the low-level high-low couplet allows the warm Pacific air to be advected into Yukon and the Mackenzie basin. On the other hand, the reduced onshore flow could decrease the precipitation along the British Columbia coast and the associated weakening of adiabatic warming and lee-cyclogenesis over the southern Prairies. These combined effects enhance the S-N anthropogenic warming gradients (Fig. 4c), weaken the background surface frontal zone, and

contribute to the development of the anomalous surface ridging in the region. For example, the mean DJF N-S near-surface temperature gradient over the southern Prairies is projected to decline by ~25% towards the end of the century. In addition, the weakened mean upper northwesterly flow over northwestern Canada reduces the number of North Pacific systems that enter Alaska to migrate down to the southern Mackenzie basin. Collectively, these changes in circulation and dynamic features are expected to reduce the frequency and intensity of weak cyclones that typically affect the region in winter.

Despite these considerations, winter precipitation at the end of the century is projected to increase by approximately 19% over the Prairies and by larger amounts at higher latitudes. This 19% increase in precipitation is slightly lower than the rate predicted by the Clausius-Clapeyron scaling (~22%, based on temperature increases over the north Pacific and Atlantic Oceans). This result can be explained by considering the anomalous upper and low-level troughs projected to occur off the U.S. west coast

which allow more moisture-laden Pacific systems to develop and affect the western U.S., as reflected in the enhanced troughing and precipitation projected over this region (Figs. 4b and d). With the weakened mean upper northwesterly flow over western Canada, some of these moisture-laden southern systems would be able to track into the Prairies and produce more frequent extreme winter precipitation events.





Preliminary analysis of daily Prairie precipitation from a 21 elements subset of the 39 CMIP5 models provides strong support for the above inference (not shown).  n particular, although the ensemble mean frequency of precipitation days (daily precipitation P > 0.5 mm d$^{-1}$) only increases marginally from 743 days during 2006-2020 to 778 days during the last 15 years of this century, the corresponding frequency of extreme precipitation days (daily P > Pc, the 99$^{th}$ percentile of daily P during DJF 2006-2020) increases from 10 to 29. In addition, although the mean frequency of extended (longer than 1 day) precipitation events hardly changes between the two periods (157 versus 158), the frequency of extended extreme precipitation events (multi-day events with daily P > Pc) increases 5-fold from 0.6 to 3.3 between these 2 periods.

## 3.3  Spring

The most prominent circulation anomaly feature is the quasi-stationary upper low centered over the northwest U.S. (Fig. 5a). This large-scale setting favours the development of cyclones that bring warm-season precipitation to southwestern Canadian regions (Szeto et al., 2011, 2015, 2016).  The intensity of the low can be quantified by the H-index as detailed in Szeto et al. (2016) and the development of this feature is shown in Fig. 6. The low is a robust feature projected by most models (Fig. 6a). The magnitude of the low (i.e., negative H-index) is projected to intensify until ~2060 when it becomes stabilized. At the lower levels, the S-N warming gradient is relatively weak (Fig. 5c), and thus has little effect on the mean frontal zone across the southern Canadian regions. As a result, cyclone activity that affects the region is expected to increase, as reflected in the anomalous N-S surface trough that extends from Hudson Bay into the eastern Prairies and central U.S. (Fig. 5b).

Consequently, spring precipitation is projected to increase significantly over southern Canada in general (Fig. 5c). For example, Prairie MAM precipitation increases by 26%. In fact, spring is the only season with projected P increase larger than the rate predicted by the Clausius-Clapeyron scaling (~21%, based on the temperature increase over the Pacific and Atlantic Oceans). This Prairie spring precipitation is projected to increase starting from the 1990s and continue until around 2060 (not shown) following the stabilization of the anomalous upper low. It is noteworthy that the predicted intensification of the upper low and associated increasing trends of mean and extreme precipitation over the eastern Prairies during the turn of the century are also evident in observations (Szeto et al., 2015).

Results from the analysis of daily Prairie precipitation provide further insight into the regional precipitation response to the circulation change. In particular, the frequency of extreme precipitation days (daily P > Pc, where Pc is the 99$^{th}$ percentile of daily P during MAM 2006-2020) doubles from 10.5 days during the early-century period to 20.4 days towards the end of the century (2086-2100). In addition, although the mean frequency of extended precipitation events hardly changes between the two periods (153 versus 156), the frequency of extended extreme precipitation events increases 3-fold from 1.2 to 3.7 between these 2 periods. Although both the DJF and MAM results suggest substantial future increases in extreme precipitation events, it is noteworthy that, although the relative seasonal precipitation increase for MAM is higher, the increase of extreme



precipitation event frequency is somewhat higher for DJF. The apparent discrepancy is likely related to the difference in the model ensembles (21 vs 39 members for daily and monthly analysis, respectively) that were used in the assessments.

The long-term mean upper low anomaly would allow more upper low systems to enter the continent through the northwest U.S. Analysis of historical extreme Prairie precipitation events (not shown) suggest that the location of heavy precipitation is sensitive to the location of the upper low due to the topography that characterizes the region. In particular, strong upslope rainstorms over southern Alberta, similar to the one that caused the 2013 Calgary flood (Liu et al., 2016A; Kochtubajda et al., 2016), could result from upper lows that are located over the northwest U.S., whereas flood producing extreme rain events over the eastern Prairies (see for example, Brimelow et al., 2014; Szeto et al., 2015) could result from upper lows that were centered only slightly to the east. Furthermore, some systems that track slowly across the region could bring extreme precipitation to both the eastern and western regions (e.g., Szeto et al., 2011). When combined with the increased winter precipitation and earlier snowmelt and freshet in a warmed climate, the expected increase in extreme spring precipitation could substantially increase the risk of extreme Prairie spring floods over both the western and eastern Prairies.

## 3.4 Summer

In contrast to the projected spring conditions, the most prominent circulation anomaly feature is the quasi-stationary upper high centered over the northwest U.S. and southern British Columbia (Fig. 7a). This blocks cyclones that bring warm-season precipitation to southwestern Canadian regions. The decreased summer cyclone activity is also reflected in the anomalous N-S surface ridge over the western continent (Fig. 7b). Schubert et al. (2016) had previously pointed out that SST values over the Pacific also affect precipitation deficits over the continent, and Li et al. (2018) indicated that expected higher SST values in the central Pacific by the end of the century will affect the Madden-Julian Oscillation in a manner conducive to reduced summer precipitation over the Canadian Prairies. It is not clear though whether the anomalous upper high is related to such pattern changes over the central Pacific.

The development of the anomalous upper high is evident in the time series of the JJA H-index (Fig. 6b). Unlike the spring upper low that is projected to be located at the same general location, the intensification of the high is expected to accelerate after ~2060-2070. Under the influence of the upper high, downward solar radiation is projected to increase up to ~5 W m$^{-2}$ over southern Prairies by 2100. JJA near-surface air temperature is expected to increase by ~2°C over the next 30 years (by 2050) and by an additional ~4°C over the following 50 years (by 2100) along with even stronger warming over its southern vicinity. As a result, a "hot spot" with maximum summer warming that extends into southwestern Canada is projected to be induced under the upper high (Fig. 7c). The projected large-scale changes induce a significant decrease in precipitation over the Great Plains (Fig. 7d). In accordance with the temporal development of the upper high, the Prairie JJA precipitation is projected to remain rather constant until approximately 2070 and then decrease by 5% towards the end of the century. Although this is not a significant decrease, summer is the only season with projected reduction in precipitation over the study region. In





contrast, evapotranspiration (not shown) is projected to increase slowly until 2060 (~6 % of historical values) and then decrease very slowly again towards the end of the century.

The reduction in summer precipitation, along with the enhanced evapotranspiration induced by the strong surface warming could increase the potential for summer drought and wildfires over western Canada. Increases in surface sensible and latent heat fluxes that would accompany the projected strong warming could also enhance convective activity. On the other hand, the projected upper high is expected to suppress convection when fully-developed. As a result, it is not clear how summer convection might change during the mid-to-late century; it may be enhanced or suppressed.

## 4  Examination of critical phenomena

Although Sect. 3 addressed numerous phenomena, three overarching categories were investigated in more detail. They are organized under cold season, spring and early summer, as well as summer issues.

### 4.1  Cold season and near 0°C conditions

The location of the 0°C isotherm is a critical aspect of this region's climate. It is closely linked with the melting of snow at the surface which in turn affects albedo and land-atmospheric energy exchange (Jennings et al., 2018). Precipitation near this temperature furthermore varies greatly in occurrence and type and can be linked with major hazards (e.g. freezing precipitation). Changes in these features are examined here.

#### 4.1.1  0°C isotherm movement and pattern

In association with overall warming, the near 0°C region will move northward. To quantify this, monthly average locations of the 0°C isotherm were calculated from different model datasets. The NCEP/NCAR re-analysis was used to compute locations for 1976-2005. The future time periods were computed by adding the CMIP5 39-model ensemble median of the RCP8.5 10-y mean air temperature delta (future period – 1976-2005 historical) projected for 3 different periods (2046-2055, 2066-2075, 2086-2095) to the NCEP/NCAR climatology.  Locations of the isotherm for each ensemble member were estimated by interpolating air temperatures onto a 1 degree by 1 degree latitude-longitude grid.

Results of these calculations are shown for two months (March and November) that illustrate some of the greatest movements (Fig. 8). In central regions of the country, the movement of the 0°C isotherm is of order 50-100 km per decade (especially in November) although it is much less in some areas of high terrain in the Western Cordillera. The high terrain means that the near 0°C region would move vertically but little horizontally.





Note that there is considerable variation between models in the actual locations of this isotherm. Some of the narrowest spreads occur in the interior of the country, far from oceans and mountains. In the western Cordillera, the spread is large in part due to different regions having high terrain which strongly influences the locations of this isotherm. Oceanic regions also exhibit large spreads with the East Coast and Hudson Bay being impacted by variable sea ice cover.

This large spread in the projected western Cordillera patterns is accentuated when considering whether rain or snow will fall. For example, in the Kananaskis area of the Alberta foothills, a mixture of rain and snow has been observed at temperatures as high as 9°C in some events, whereas it only occurred below 2-3°C in others (Thériault et al., 2018). Critical factors behind such varying observations are the atmospheric moisture content and the density of the falling solid precipitation. For example,

high (low) values of moisture content and low (high) values of particle density lead to rapid (slow) melting and a low (high) upper temperature threshold. However, particle density may be high if the moisture content is as well due to a greater likelihood of supercooled droplets aloft that can accrete onto solid particles. Such factors including their inter-connections make the determination of the rain-snow transition a challenging issue even in the present climate, let alone the future one.

The character of the near 0°C regions over generally flat terrain will also change. Surface observations and the WRF CONUS dataset (described in Sect. 2) were examined at several locations across the domain to illustrate these changes. An example is Winnipeg, Manitoba (Fig. 9). As shown by the observations, the annual cycle of its near 0°C conditions (temperature in the range - $1°C \leq T \leq 1°C$) is characterized by no occurrences in summer and only a few mid-winter. But there is considerable variability including onset times in the autumn, cessation times in the spring, and the overall annual number. Precipitation can

be linked with the events and, when present, it tends to occur towards the 'winter' side as opposed to the 'summer' side. A similar comment applies to the timing of the top 5% in terms of duration. The 'summer' side events are mainly linked with the diurnal cycle with daytime temperatures above 0°C and nighttime ones below; temperatures pass quickly through 0°C. In contrast, events occurring on the 'winter' side tend to be linked with frontal systems and therefore are associated with precipitation. Temperatures may rise to just above 0°C. Such conditions would also melt some of the snowpack when

temperatures are above 0°C and freeze melt water when below; both processes act to maintain temperatures near 0°C through, respectively, cooling and heating by the latent heat of fusion. WRF CTRL tended to be somewhat warmer (colder) in the warm (cold) months than the observations, as discussed by Liu et al. (2016A). This led to fewer near 0°C regions on the 'summer' side of the model's annual cycle.

The annual cycle of near 0°C conditions is projected to undergo change. As expected, there will be a longer period in the summer without such conditions under PGW; such an increase has been well documented across Canada (Bonsal and Prowse, 2003). From the WRF CTRL to PGW, the last day in the spring with near 0°C conditions occurred earlier by an average of 16 days, and first day in the autumn with near 0°C conditions occurred later by an average of 18 days. In contrast, transition regions occurred more frequently in mid-winter. WRF CTRL simulated too few hours near 0°C (4606 rather than 5623



observed) but WRF PGW simulated 5372 hours. Consequently, the projected future number of hours of occurrence actually increases, although its maximum duration does not change. Precipitation infrequently occurred with near 0°C conditions (18% and 21% of events in the observations and WRF CTRL, respectively) but this fraction increased substantially (28%) in the moister PGW projections. When precipitation did occur, it continued to be mainly on the 'winter' side of the seasonal

distribution and, in these precipitation events, the fractional occurrence of the different types of precipitation was relatively constant with snow always dominating (approximately 60% of occurrences).

### 4.1.2 Freezing rain

The movement of near 0°C conditions must be linked with changes in freezing rain occurrence. This type of precipitation, not even considering its accumulation, is difficult to simulate and project into the future. Atmospheric factors driving its formation

over the CCRN region even include chinooks (Kochtubajda et al., 2017a); surface factors include the degree of sea ice over Hudson Bay that, when present, acts to maintain the necessary cold near-surface temperatures. The generally low occurrence of freezing precipitation, in comparison with other regions, is partially attributable to particle sublimation or evaporation below cloud (Kochtubajda et al., 2017a).

To assess future changes in the occurrence of freezing rain, the fifth generation of the Canadian Regional Climate (CRCM5) model with a 0.44° grid mesh was used (Fig. 10). CRCM5 was driven by the GCM from the Max-Planck Institute for Meteorology Earth System Model (MPI-ESM-MR) for the 1981-2000 and 2081-2100 periods using the RCP8.5 scenario (Moss et al., 2010). Freezing rain was diagnosed using the technique developed by Bourgouin (2000); this approach is used operationally at ECCC. Results shown in Fig. 10a indicate that the model driven by MPI-ESM-MR reproduced the general

pattern of the mean annual number of hours of freezing rain over the domain and basically shows the same features as a hindcast simulation driven by reanalysis data (not shown) but somewhat weaker. The main differences are a strong negative bias west of the Hudson Bay as well as in an area stretching from northern Alberta to southern Manitoba. This latter area is likely due to the cold bias for that region as suggested by Šeparović et al. (2013); it is produced by the cold sea surface temperature bias in Northern Pacific in the driving data.

Projections suggest little change in the southern portion of the region but increases in excess of 20 h yr$^{-1}$ in the Northwest Territories. Such increases are comparable to current annual values. This pronounced increase in the north is due, at least in part, to the northward movement of near 0°C temperatures (Sect. 4.2.1).

Occurrence patterns of freezing rain over the annual cycle have also been organized into regimes (Kochtubajda et al., 2017a) and their locations may also change. Five regimes are possible and range from too cold everywhere (regime 1) to too warm everywhere (regime 5). Over the CCRN region, almost all locations are in regime 4 (freezing rain does not occur during summer but can in any other month) but higher latitudinal locations are either in regime 2 (only occurs in summer) or region



(does not occur in at least one summer and one mid-winter month). With climate warming, these northern locations may shift into higher-numbered regimes. For example, at Resolute, Nunavut at 74.7°N (regime 2), freezing rain only occurred during 5, 1-hour events in July over a 42-year period (1964-2005) and these were associated with subfreezing surface temperatures ≥ -1.8°C. If these temperatures had just been > 0°C, rain would have been reported and Resolute would be in
regime 3.

## 4.2 Spring and early summer flooding

Flooding often occurs across this region in the spring and early summer. Two such areas are in the western and eastern Prairies and each has experienced devastating events recently. Future aspects of these floods are examined here from an atmospheric perspective.

### 4.2.1 Alberta upslope precipitation and flooding

As described in Sect. 3, the enhancement of troughing over the northwest U.S. is critical for localized precipitation in southern Alberta. This applies to the catastrophic flooding that occurred in June 2013 (Pomeroy et al., 2015; Liu et al., 2016A; Kochtubajda et al., 2016; Li et al., 2017). This event arose in large part through numerous atmospheric factors ranging from the large scale conditions down to precipitation distributions over the affected basins.

The location of the anomalous upper low in this event was similar to those associated with observed extreme rainstorms over the western Prairie (Szeto et al., 2011). As such, at least some of the extended extreme precipitation events that are projected to occur during MAM (Sect. 3.3) could be expected to be associated with precipitation extremes over the western Prairies. Although coarse-grid CMIP5 models are not able to accurately simulate the complex feedback processes that are critical in
development of such storms (Szeto et al., 2011), such events have been simulated successfully with high resolution cloud-resolving models such as the WRF (Li et al., 2017). Hence an analysis of WRF PGW information to examine how the characteristics of these storms might change in a warmed climate is justified and warranted.

The WRF CONUS information is shown in Fig. 11. Temperatures increased from the CTRL to the PGW simulations by an
average of 4.2°C. Over the CCRN domain, increases were most dramatic in the extreme southern part of Alberta (approximately 6°C) as well as along the continental divide (approximately 5°C). The former warming would have led to a sharpening of the surface front across southern Alberta (Liu et al., 2016A) with regions to its south warming approximately 2°C more than over regions to its north. Although the large scale forcing was unchanged under PGW assumption, the embedded frontal structure was enhanced.

Precipitation increased by 7.7% over the domain shown in Fig. 11. But, there were wide variations with some regions experiencing a substantial decrease (> 50 mm). The enhanced frontal region over southern Alberta was associated with greater



precipitation but both increases and decreases occurred along the continental divide to its west. Both the CTRL and PGW simulations reproduced the front-parallel intense convective precipitation bands that occur just ahead of the surface warm front that were observed in Liu et al. (2016A) and Kochtubajda et al. (2016). As evident in Fig. 11, the precipitation enhancement in PGW is largely associated with the stronger frontal convective bands that were simulated in the warmed environment.

Although the detailed involved mechanisms are not clear yet, these results suggest that the atmosphere's response to frontogenesis was enhanced with the imposed large-scale atmospheric warming and moisture increase to induce mutual-amplifications of both frontogenesis and frontal precipitation development.

Although there were substantial variations under the warmer and moister environment, it is important to determine whether

similar features were associated with the production of large amounts of precipitation. The locations of precipitation maxima in the observations, CTRL simulation and PGW simulation were all in the Alberta foothills east of the continental divide. The CTRL maximum value was not at the observed Burns Creek location (Kochtubajda et al., 2016) but it was only 16 km away (to the southeast); the PGW maximum value was still just 50 km away (to the northwest). The temporal evolution of precipitation at these locations was also similar; precipitation initially occurred at high rates followed by long-lasting, lower

values. At these locations, precipitation at the surface was always rain, the peak in precipitation rate was always linked with graupel as well as snow aloft, and snow aloft dominated at all other times. The peaks in precipitation rate were also associated with the highest values of CAPE (convective available potential energy).

The PGW results suggest that the severity and impacts of western Prairie floods that are produced under similar synoptic

conditions could be substantially increased under climate change although similar patterns are associated with the greatest precipitation in the Alberta foothills. Due to the significance of such implications, further studies are warranted of the attendant storm-scale processes that were involved in producing the results, and whether more sophisticated modeling approaches than PWG are required to better simulate the impacts of warming on such and similar storms.

### 4.2.2   Eastern Prairie floods

Large to synoptic scale atmospheric forcing is critical to the likelihood of spring and early summer flooding over the eastern Prairies. In particular, persistent atmospheric patterns often bring extended periods of precipitation extremes, either wet or dry, depending on location across the region relative to the circulation pattern (Brimelow et al., 2014; Brimelow et al., 2015; Szeto et al., 2015). Such persistent patterns were linked with spring or early summer rainfall that contributed to flooding on the Assiniboine River in 2011 and 2014; this precipitation also coincided with snow melt.

A few studies have examined how such persistent patterns are expected to change across different regions of the Prairies. Szeto et al. (2015) found that persistent patterns conducive to eastern Prairie spring and early summer enhanced precipitation and flooding may become more pronounced. Using 500 hPa output from several RCM/GCM combinations, Bonsal et al. (2017)





and Bonsal and Cuell (2017) identified future (2041-2070) changes to the frequency of key summer (JJA) circulation patterns associated with extreme dry and wet conditions over the south-western Canadian Prairies and the Athabasca River Basin (ARB), respectively. Most of the models simulated general features of observed circulation patterns, and these also occur in the future but with some changes to their average frequency. However, there was considerable inter-model variability.

Flooding events in Canada are often associated with numerous factors (including extreme precipitation) that occur in combination. For example, one critical aspect of the 2014 Assiniboine flood was a cool spring followed by rapid snowmelt combined with above normal precipitation (Szeto et al., 2015). MAM temperatures were approximately 2.5°C below the 1995-2014 normal and AMJ precipitation was approximately equal to the 90[th] percentile of AMJ precipitation that occurred during
10   1950-2005.

CMIP5 information was examined to determine whether the likelihood of this combination would change in the future. Results in Fig. 8a show that spring melt would commence in February and be completed in March towards the end of the century. As such, the frequency of wet MAM and cool FM during 2081-2100 is compared to historical (1986-2005) wet AMJ combined
15   with cool MA, using the anomalous conditions for the 2014 flood as criteria for each model. The focus area is the eastern Prairies (47.5-55°N latitude by 255-262.5°E longitude).

Results indicate that the ensemble mean frequency of wet MAM (i.e., MAM with P> 90[th] percentile of MAM precipitation during 1950-2005) is projected to be 4.6/decade, which is substantially higher than the corresponding mean frequency of
20   1.2/decade for wet AMJ during 1986-2005. In addition, the mean frequency of combined wet MAM and cool FM (i.e., FM temperature 2.5°C cooler than the 2081-2100 mean) was projected to be 0.8/decade, which is 8 times larger than the frequency of 0.1/decade estimated for the co-occurrence of wet AMJ and cool MA during 1986-2005. The results suggest that the projected large-scale atmospheric conditions that are favorable for the development of wet MAM could substantially increase the risk for eastern Prairie floods that are associated with cool and wet springs.

### 4.3 Summer severe conditions

Summer across the region is typically linked with severe conditions. These range from widespread drought to severe thunderstorms. Their combination can furthermore be key factors linked with forest fires. Expected changes in these phenomena are examined here.



### 4.3.1 Drought

The projected large-scale summer changes (Fig. 7) are expected to impact future drought conditions. Given that past droughts over western Canada have been associated with a persistent mid-tropospheric (500 hPa) large-amplitude ridge centered over the area (e.g., Bonsal et al., 1999), it is anticipated that the quasi-stationary anomalous upper high centered over the northwest

U.S. and southern British Columbia (Fig. 7a) will result in more drought-like conditions in this region. Some evidence for these changes was found by Bonsal et al. (2017) and Bonsal and Cuell (2017) who examined future (2041-2070) changes to summer (JJA) Standardized Precipitation Evapotranspiration Index (SPEI, Vicente-Serrano et al., 2010) values over two southern Canadian Prairie watersheds and the Athabasca River Basin (ARB), respectively. For the southern basins, results indicated an uncertain future ranging from a substantial increase in drought, with a higher degree of inter-annual variability,

to relatively no change from current conditions. Farther north in the ARB, projections revealed an average change toward more drought-like summer conditions, but there was a substantial range among the climate models. Over a larger study area that included all western Canadian river basins, Dibike et al. (2018) incorporated six CMIP5 GCMs to assess future SPEI changes on annual and summer scales for the periods 2041-2070 and 2071-2100 (relative to 1971-2000) using RCP4.5 and RCP8.5 emission scenarios. They found that southern watersheds showed a gradual increase in annual water deficit throughout the

21st century whereas the opposite was true for northern basins. In contrast, for summer, all river basins with the exception of the extreme northern ones were expected to experience decreasing water availability.

A comprehensive Canada wide drought study assessed changes in the SPEI using outputs from 29 CMIP5 models (Tam et al., 2018). In agreement with Fig. 7, results showed strong relative summer drying during the 21[st] century over much of western

Canada including interior southern British Columbia, as well as west-central portions of the country from the Prairies to the Arctic. In addition, the frequency of extended relatively dry periods (e.g., consecutive years that are characterized by strongly negative summer SPEI) is projected to increase markedly during the second half of this century. Compared to other locations in Canada, the southern Prairies exhibit the largest likelihood of extended severe drought during the latter part of this century under the RCP8.5 scenario (Fig. 12a). The intensification, following approximately 2050, is consistent with the accelerated

intensification of the upper ridge during the second half of the century (Fig. 7a).

On annual scales, a dry-south–wet-north pattern characterizes projected drought changes over the CCRN domain (see Fig. 12b for drying over the south) (Tam et al., 2018). This pattern is largely accounted for by the combined results of projected dry conditions during the summer and autumn over southern regions, and the projected wet conditions during winter and spring

over the northern and coastal regions of Canada. The projected surface water deficit during summer and autumn would thus play a dominant role in affecting the future annual water budget over the southern domain.





An important issue concerning drought is its future intra-seasonal character. Many droughts tend to be hot with almost no precipitation. In contrast, others are not associated with especially elevated temperatures and can even have cool periods (Stewart et al., 2012), can have rain showers (Evans et al., 2011), and/or experience occasional large precipitation events (Szeto at al., 2011) that may or may not increase in a warmer world (as discussed in Sect. 4.2.2). Such differences in character can

substantially affect their impact but the CMIP5 information does not have sufficient resolution to resolve this issue.

### 4.3.2  Convection and hail

The analyses have so far largely examined expected changes by the end of the century. But many of these tend to become more pronounced with time. An important issue is to examine change by mid-century when, as discussed in Sect. 3.4, dry conditions are not expected to be so dominant over the southern Prairies (Fig. 12).

To address this, the NARCCAP historic (1971 to 2000) and mid-century future (2041 to 2070) model output (Mearns et al., 2013) was used to assess future changes in hail as well as total and convective precipitation over the Canadian Prairies, southern Northwest Territories and U.S. northern plains. Convective precipitation is defined to occur when the model convective scheme is triggered to release latent energy and convective instability through simulated vertical motion and total precipitation is the

sum of non-convective and convective precipitation production processes. Brimelow et al. (2017) suggested that the three most consistent NARCCAP model pairings to assess convective precipitation and hail for the regions of interest herein, included MM5-HadCM3, MM5-CCSM and HRM3-HadCM3, based on their ability to reproduce the precipitation climatology.

Changes in future warm season (JJA) total precipitation are shown in Fig. 13 for three NARCCAP model pairs using the

Special Report on Emissions Scenarios (SRES) A2 scenario. Consistent increases (10-60 mm) with all three model pairs occur in central and northern Manitoba as well as west central and most of northern Alberta. All model pairs show little to no change in total precipitation in central to northern Saskatchewan. Two model pairs (MM5-HadCM3 and MM5-CCSM) show some increases (up to 50 mm) in the northern U.S. that can include southern Saskatchewan and Manitoba, whereas MM5-CCSM and HRM3-HadCM3 show slight decreases (10-30 mm) in the southeast Northwest Territories and extreme southwest

Nunavut. Inconsistent changes between the three model pairs occur elsewhere in the domain. These results are broadly consistent with other research over the region of interest (e.g. Mailhot et al., 2011; Mladjic et al., 2011). It is not known why the MM5-HadCM3 shows large increases over Lake Winnipeg and western Lake Superior; this could be due to an overactive convective parameterization and/or evaporation scheme, since this also appears in the convective precipitation field, shown next. It is of interest to note that only the precipitation change pattern in HRM3-HadCM3 is comparable to the one generated

with CMIP5 RCP8.5 results for the same future period (not shown).

Changes in future warm season (JJA) convective precipitation are shown in Fig. 14 for three NARCCAP model pairs under the SRES A2 scenario. All three model pairs show increases (10-70 mm) in several areas of the Prairies, but with little to no





change in central Saskatchewan. MM5-CCSM and MM5-HadCM3 show the most dramatic increases, whereas the HRM3-HadCM3 shows the weakest; it is not known why, although it is likely due to differences in how convection is treated. The MM5 model pairs also show consistent increases (10-60 mm) in the southern Northwest Territories and across the U.S. northern plains while the HRM3-HadCM3 has little to no change in convective precipitation in those same areas. These results are consistent with general increases in CAPE and surface dew points in a warming climate scenario over much of the Prairies (Brimelow et al., 2017).

The future occurrence of hail is also important. A recent study by Brimelow et al. (2017) highlighted future changes (2041–2070 minus 1971–2000) in hail character over North America based on simulations from a one-dimensional cloud-hail model (HAILCAST) forced with the same three NARCCAP model pairs discussed above (Mearns et al., 2012). Over the CCRN domain, results show that the number of hail days generally decline in Manitoba and Saskatchewan in summer (JJA), while increases occur in the western half of Alberta and extreme southwest Northwest Territories (see Brimelow et al., 2017 Fig 1). Over much of Alberta and southwest Northwest Territories, there are general increases in accumulated kinetic energy (AKE) and maximum hail size, even in some regions where the number of hail days does not change in the future (see their Fig 2). That is, when it does hail, it will potentially be larger and more destructive (larger AKE). Parts of Saskatchewan and central to northern Manitoba may also see increases in AKE and hail size even though the number of hail days decreases. This is thought to be primarily due to more moisture and energy available to summer storms when they do occur (Brimelow et al., 2017).

A more detailed analysis of summer (JJA) future hail changes was carried out for the major cities on the Prairies (Winnipeg, Regina, Saskatoon, Calgary and Edmonton) using the same dataset as Brimelow et al. (2017). Changes in occurrence frequency versus hail size were examined for each city (no figures are shown). Several model grid points were used to represent each city within a 50 km radius of the city's latitude-longitude center. Results are broadly consistent with the findings of Brimelow et al. (2017). No statistically significant (with 90% confidence) changes in hail character (median hail diameter and median AKE) take place in Winnipeg and Regina, based on the Wilcoxon Rank Sum Test, whereas significant decreases in median hail diameter (by 1 cm) and median AKE (by 1-2 J) occur in Saskatoon. General increases in median hail diameter (by 0.5 - 1 cm) occur in Edmonton and Calgary, however, changes in AKE are mixed depending on the NARCCAP model pair used for each city.

Overall, the mid-century results from these NARCCAP regional model studies are largely consistent with the earlier CMIP 5 discussion (Sect. 3.4) on the evolution of convection over the region. Parts of the Canadian Prairies may see slightly increased total precipitation with much of this due to more convective activity, however, parts of Saskatchewan may experience no change or slightly less total precipitation despite no change or slightly greater convective precipitation.




### 4.3.3 Lightning

As indicated in Sect. 3.4, convection may be enhanced or suppressed by the end of the century. A related issue is lightning. Since long term observations by satellite-based or ground-based lightning location systems of lightning do not exist, studies assessing past trends have used thunderstorm day records. Trends have varied around the world (Changnon and Changnon, 5   2001; Pinto et al., 2013). To our knowledge only one such study has been carried out within Canada. Huryn et al. (2016) examined nine weather stations across southern Ontario and reported significant trends in thunderstorm occurrence at four of the stations. No research has yet been done over the CCRN region in terms of past trends in days with lightning.

Several studies and climate model simulations have reported lightning activity increases in a warmer climate. For example, 10   Price and Rind (1994a,b) used the Goddard Institute for Space Studies general circulation model to estimate the effect of a $2xCO_2$ climate on global lightning. They used a power function of cloud-top height as a proxy for lightning activity and reported an approximate 5-6% change in global lightning for every 1°C of temperature change. Michalon et al. (1999) subsequently extended the Price and Rind (1994a) cloud-top height parameterization by taking into account the role of cloud droplet concentrations on lightning. Their $CO_2$ doubling simulations suggested a 10% increase in global lightning for a surface 15   warming of 2°C. Romps et al. (2014) used the product of the precipitation rate and the convective available potential energy as a proxy for lightning flash rates to assess how future warming will affect lightning over the contiguous U.S. An ensemble of 11 GCMs predicted an increase in lightning strikes at a rate of 12 ± 5% for every 1°C of warming. Based on Romps et al. (2014), IPCC (2014) pointed out an expected increase in the occurrence of extreme events could include lightning.

20   Recently, an upward cloud ice flux parameterization was added to a chemistry-climate model to simulate future lightning in 2100 under the RCP8.5 scenario (Finney et al., 2018). Comparisons to the cloud-top height parameterization (Price and Rind, 1994a) were also made. Globally, the ice flux scheme projected a decrease in lightning activity largely due to a large reduction in the tropics. This is in contrast with the cloud-top height scheme which projected a global increase. Over the CCRN region, the ice flux scheme projected an increase at latitudes above approximately 60°N, in agreement with Price and Rind (1994a), 25   but a decrease of 0.1 - 10 fl km$^{-2}$ yr$^{-1}$ (not statistically significant) over some parts of the Prairies which differs from Price and Rind (1994a).

Studies examining future lightning have not addressed polarity. Generally, an overwhelming majority of strokes are negative although this is not always the case (Logan, 2018). Any alteration in the relative distribution of polarity can have a major 30   impact on the likelihood of wildfire initiation with positive strokes generally leading to a greater likelihood of fire ignition. Several studies have suggested that wildfire smoke can actually alter the microphysical factors governing precipitation and lightning production processes resulting in and an increased fraction of positive cloud-to-ground flashes (Lyons et al., 1998; Murray et al., 2000; Fernandes et al., 2006; Altaratz et al., 2010; Kochtubajda et al., 2011).

Lightning occurring with relatively little or no accompanying rainfall, known as dry lightning (Rorig and Ferguson, 1999), is of particular concern as an ignition source for wildfires. Such events arise from thunderstorms with, for example, high cloud bases and dry sub-cloud conditions which lead to the evaporation of the falling precipitation (virga). One study

combined lightning observations with atmospheric conditions to specify large-scale environments favourable to its occurrence in Australia (Dowdy, 2015). This technique was then applied to GCM scenarios to examine future dry lightning activity over Australia; results suggest considerable seasonal and spatial variability of the projected changes in environments favorable to dry lightning occurrence.

### 4.3.4  Wildfires

Fire activity is influenced by three factors; fuels, ignition sources, and weather conditions (Flannigan and Wotton, 2001). The number of wildfires and areas burned over the CCRN region has varied dramatically from year-to-year.  For example, the Northwest Territories on average experiences 279 fires, which consume nearly 5,700 km$^2$ annually. During 2014, a record 33,900 km$^2$ burned (Kochtubajda et al., 2019). As well, in early May 2016, Canada's costliest natural disaster and Alberta's third largest fire event in its history occurred around Fort McMurray. A rare feature reported during this event was that lightning

from a pyrocumulonimbus cloud ignited four fires (Kochtubajda et al., 2017b).

The findings described in previous sections of this article are consistent with an increase in wildfires in the future. Higher temperatures and more drought are expected. More lightning may also occur but little is known about some of its future features that are linked with wildfire ignition.

Overall, the likelihood of more wildfires is consistent with previous studies. For example, Flannigan et al. (2005) and Flannigan et al. (2009) expected that wildfires will increase throughout the Northern Hemisphere in a warmer climate. Mann et al. (2017) pointed out that, as the future climate warms, northern latitudes are projected to experience greater persistence of large scale circulation patterns that can be conducive to wildfires. Wildfire occurrence and its future impacts on terrestrial ecosystems,

cryosphere, and hydrological functioning are addressed in more detail in part two of this review and synthesis.

### 5  Synthesis of future conditions

The preceding information has examined large scale expected seasonal change as well as its impact on smaller-scale events with a focus on physical processes and inter-connections. The basis for this insight rested on new research findings as well as published articles. This insight is pulled together into conceptual frameworks largely applicable to the end of the century (Fig.

30 15).



Based on this insight, the future climate is expected to include substantial change. This includes strong and distinct seasonal dependence of large-scale dynamic drivers and a general increase in 'intensity' of these drivers. Upper level and surface patterns sometimes conspire, for example, to increase cyclonic activity but reduce it in other seasons. This overall setting is expected to have major impacts on regional and local scales. These include patterns in hydroclimatic responses that vary with

season. In particular, the expectation is for greater excesses and deficits of precipitation as well as its intensity and character. There will also be distinct shifts in events directly related with temperature including those near 0°C.

These expected changes can be summarized seasonally. In autumn, the projected upper air circulation change resembles a westward-shift negative PNA pattern that leads to more frequent but generally weaker frontal cyclones, and associated

increases in precipitation and freezing rain, over the southern CCRN region. In contrast, upper circulation change that resembles an eastward-shifted positive PNA pattern is projected for winter. The frequency of weak winter cyclones would be reduced but more intense, major snow storms over the southern CCRN region is expected. In spring, a pronounced upper low anomaly just southwest of the CCRN region will be conducive to more cyclonic systems and precipitation and more likelihood of spring flooding. In summer, a pronounced upper level high pressure anomaly to the southwest of the CCRN region will be

linked with a greater likelihood of somewhat decreased precipitation as well as drought and forest fires.

Information on the timing of change is critical for the development of effective mitigation and adaptation measures. Analyses of the time evolution of regional hydroclimate responses show that the development of many hydroclimate variables and extremes (e.g., extreme Prairie drought) is projected to be accelerated near mid-century. The new insights gained on the

physical relationships between the large-scale and regional responses to climate change allow us to link the accelerated mid-century regional changes to the corresponding temporal behavior of the upper air large-scale drivers.

Additional and more comprehensive investigations on the evolution of changes are certainly required. For example, as shown in Sect. 4.3.2, convective precipitation may increase by mid-century before large scale circulation changes become more

prominent. It is unclear how summer convection will change, particularly by the end of the century; competing factors will be acting to enhance and suppress it. The pseudo-global warming analyses provided insights on some aspects of this issue. In particular, the results suggest that certain organized convective systems, such as the one that produced the devastating 2013 Calgary flood, might become more intense in the warmer and moister climate projected for the end of the century.

The PGW results also provide new insights on winter changes. The continuous 'above 0°C period' of the year will increase although the total period of time near 0°C may not necessarily decrease. Coupled with increased moisture, more near 0°C events will be associated with precipitation, such as freezing rain, that is often hazardous.



## 6 Concluding remarks

This article has addressed changes in atmospheric-related phenomena. The atmosphere and associated features have changed and will continue to do so due to anthropogenic factors. This certainly applies to the rapidly changing interior of western and northern Canada, the focus of the Changing Cold Regions Network (CCRN).

This study has examined conditions mainly applicable towards the end of the century over the CCRN domain and placed these within a strong physical basis. Although not as extensive, some attention was paid to the development of these conditions and pseudo-global warming analyses were carried out to examine some of the impacts of a warmer, more moist environment.

These analyses led to the development of a physically-based conceptual framework linking large scale atmospheric change to smaller scale associated features. Because of projected dramatic seasonal shifts in circulations and temperature, four conceptual models were developed to account for changes in associated phenomena. As well, insights from the pseudo-global warming analysis were brought together in a consistent manner.

Although these syntheses are based on solid physical interpretation, they have limitations. In particular, just a few scenarios or ensemble means were generally used in the analyses but there is a range of possible ones and, within each, there are numerous model products. The analysis of this information also did not carry out a comprehensive examination of the evolving pattern of large scale upper atmospheric and surface drivers; it mainly focused on their multi-year smoothed characteristics except to show their tendency to become more pronounced with time. There may well be 'surprises' or differences when these
evolving patterns are considered including those linked with flips in circulation patterns with season.

In addition, the analyses mainly relied on coarse resolution model outputs. These CMIP5 models may not properly account for all critical processes in the atmosphere, surface and boundary layer; their projections may lead to different hydroclimatic conditions than those from finer-resolution regional models. For example, numerous feedbacks from the evolving surface,
including snowcover, need to be better accounted for; these affect atmospheric circulations, storms and precipitation distributions. Other surface-related feedbacks include shifting oceanic circulations and sea ice evolution. Another key issue is ensuring that vertical atmospheric profiles are well handled over this evolving cold climate region; this is critical for atmospheric stability considerations which influence many atmospheric phenomena including precipitation distributions. Finer-resolution model outputs were only available for limited periods and did not fully account for changing large scale
circulation change. Progress made here is therefore an important accomplishment that future studies can build on.

In summary, an assessment of future weather and climate conditions over the interior of western and northern Canada has been carried out largely based on CCRN-related research. Expectations are for a future with distinct seasonal changes in large scale



atmospheric forcing, as well as temperature, and these are linked at least in part with changes in a host of associated smaller scale atmospheric-related phenomena.

Part 2 of this review and synthesis explores the associated changes at the surface and the responses and feedbacks to future
climate of terrestrial ecosystems, the cryosphere, and regional hydrology.

**Data availability:**

- CMIP5 data can be obtained from its data portals at https://cmip.llnl.gov/cmip5/data_getting_started.html (registration
required).
- The WRF HRCONUS dataset is available through the following URL: https://rda.ucar.edu/datasets/ds612.0/ (registration required).
- The NARCCAP dataset is available at: http://www.narccap.ucar.edu/data/index.html.
- The SPEI data can be obtained from CCDS (Canadian Climate Data and Scenarios) at http://climate-scenarios.canada.ca
- The NCEP/NCAR re-analysis dataset is available at:
- https://www.esrl.noaa.gov/psd/data/gridded/data.ncep.reanalysis.html
- The CRCM5 dataset is available upon request to Julie Thériault (theriault.julie@uqam.ca) and Katja Winger (winger.katja@uqam.ca).

**Author Contribution:** RES was the lead author and carried out some of the analyses. KKS carried out several analyses and wrote sections of the manuscript. BRB, JMH, BK, YL, JMT and CMD carried out analyses and contributed to the manuscript. JAB, SM and DM, PYT, ZLi and ZLiu carried out computations and contributed to the manuscript. All authors contributed scientifically by providing comments and suggestions.

**Competing interests:** Authors, John M. Hanesiak, Yanping Li, and Chris M. DeBeer, are guest editors of the special issue "Understanding and predicting Earth system and hydrological change in cold regions".

**Special issue statement:** This article is part of the special issue "Understanding and predicting Earth system and hydrological change in cold regions". It is not associated with a conference.

**Acknowledgements:** This research was supported by the Changing Cold Regions Network, funded by the Natural Sciences and Engineering Research Council of Canada (NSERC). This research was also supported by Environment and Climate Change Canada and by the NSERC Discovery grants of Ronald E. Stewart, Julie M. Theriault, John M. Hanesiak and Yanping

Li.  The authors would like to thank the National Center for Atmospheric Research for providing the WRF HRCONUS dataset that was used in this article.  The authors would also like to thank Centre pour l'Étude et la Simulation du Climat à l'Échelle Régionale (ESCER) of the Université du Québec à Montréal (UQAM) for providing the outputs of CRCM5 simulations used in this study. We specifically thank Katja Winger who provided information and output files from CRCM5. The simulations were carried out using Compute Canada facilities.

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



**Figures**

Figure 1: The region of concern for this article with the main focus being in the central region from Alberta to Manitoba and northwards to the Arctic Ocean. The names of Canadian provinces and territories, several cities, large water bodies and several land cover-related areas are also shown. The insert highlights the Mackenzie and Saskatchewan River basins.





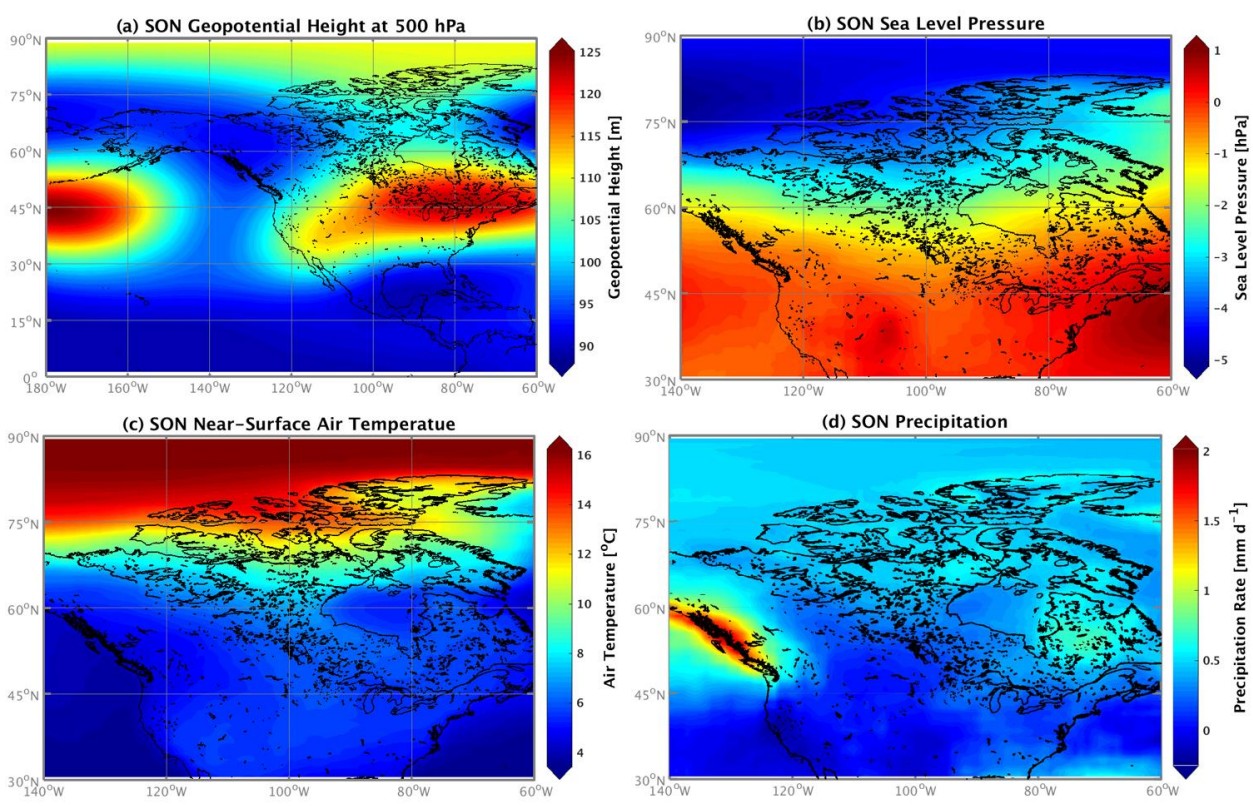

Figure 2: CMIP5 RCP8.5 projected changes of 39-model ensemble mean between the period (2081-2099) and (1981-2000) for autumn (SON). For each 3-month period, the four panels show differences in (a) 500 hPa height (m), (b) sea level pressure (hPa), (c) near-surface (2 m) air temperature (°C) and (d) precipitation rate (mm d$^{-1}$).




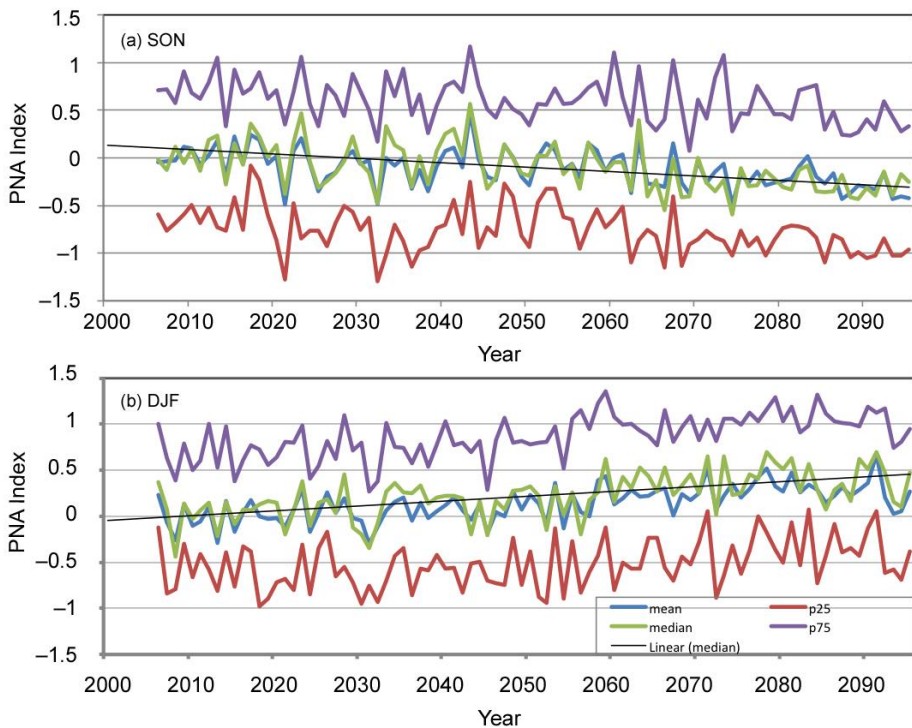

Figure 3: The (a) autumn and (b) winter PNA index over the 21$^{st}$ century computed by applying CMIP5 ensemble model information to the formula given in Wallace and Gutzler (1981). The lines indicate the mean, median, 25$^{th}$ percentile and 75$^{th}$ percentile. The linear regression for the median is also shown.





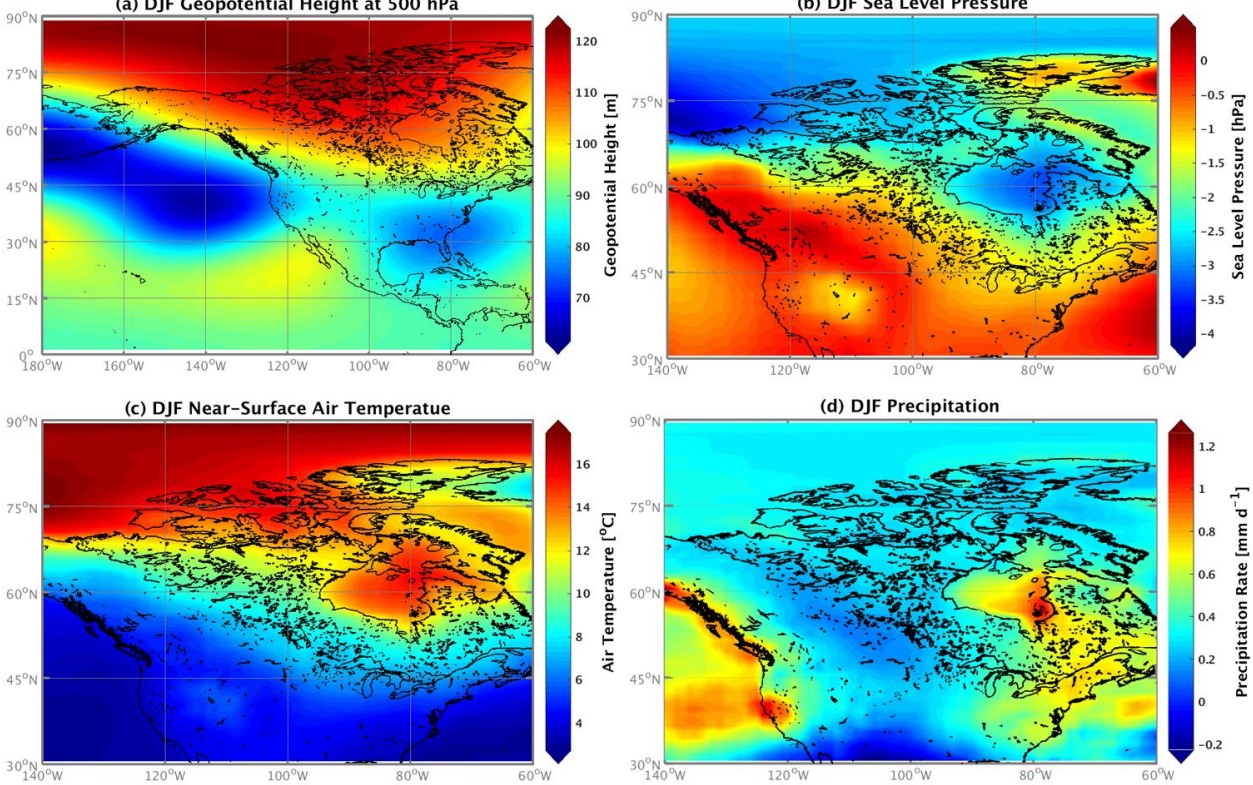

Figure 4: As in Fig. 2 but for winter (DJF).



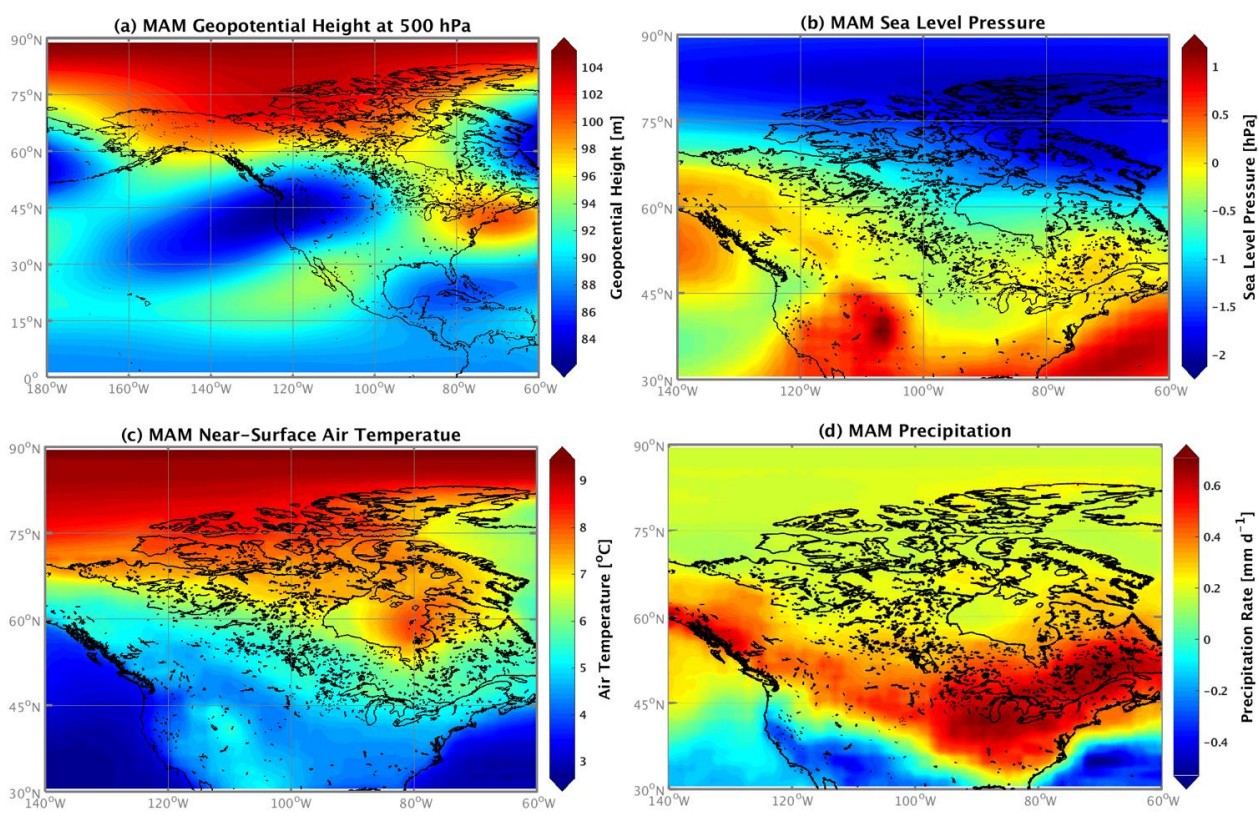

Figure 5: As in Fig. 2 but for spring (MAM).





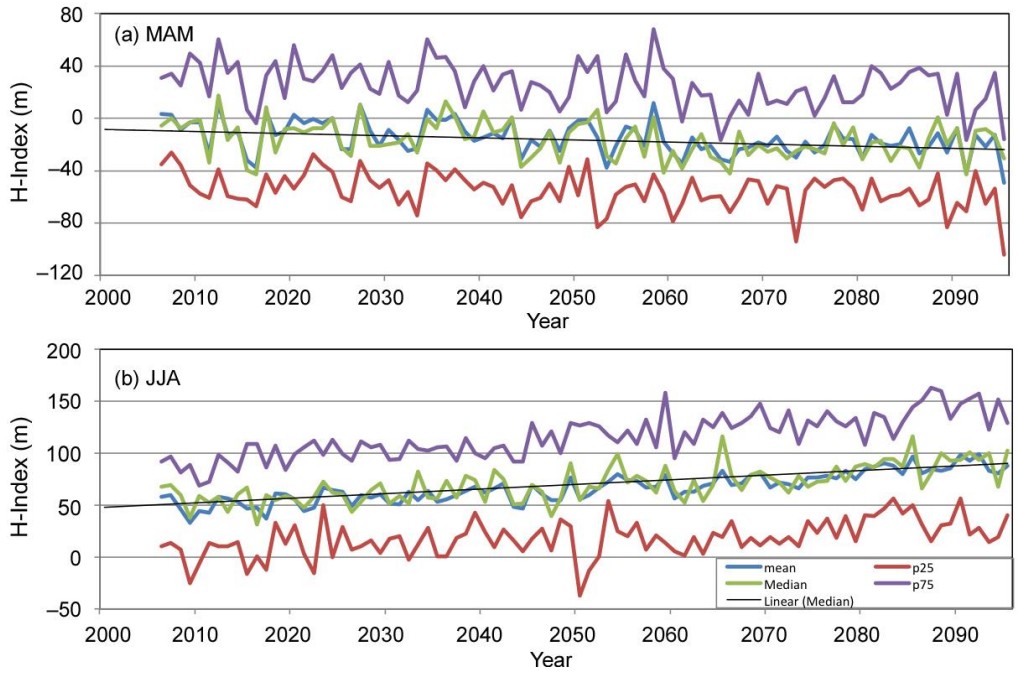

Figure 6: H-index for (a) the MAM anomalous low and (b) the JJA anomalous high centered over northwest USA during the 21st century projected from CMIP5 ensemble model information. The lines indicate the mean, median, 25th percentile and 75th

10    percentile. The linear regression for the median is also shown.





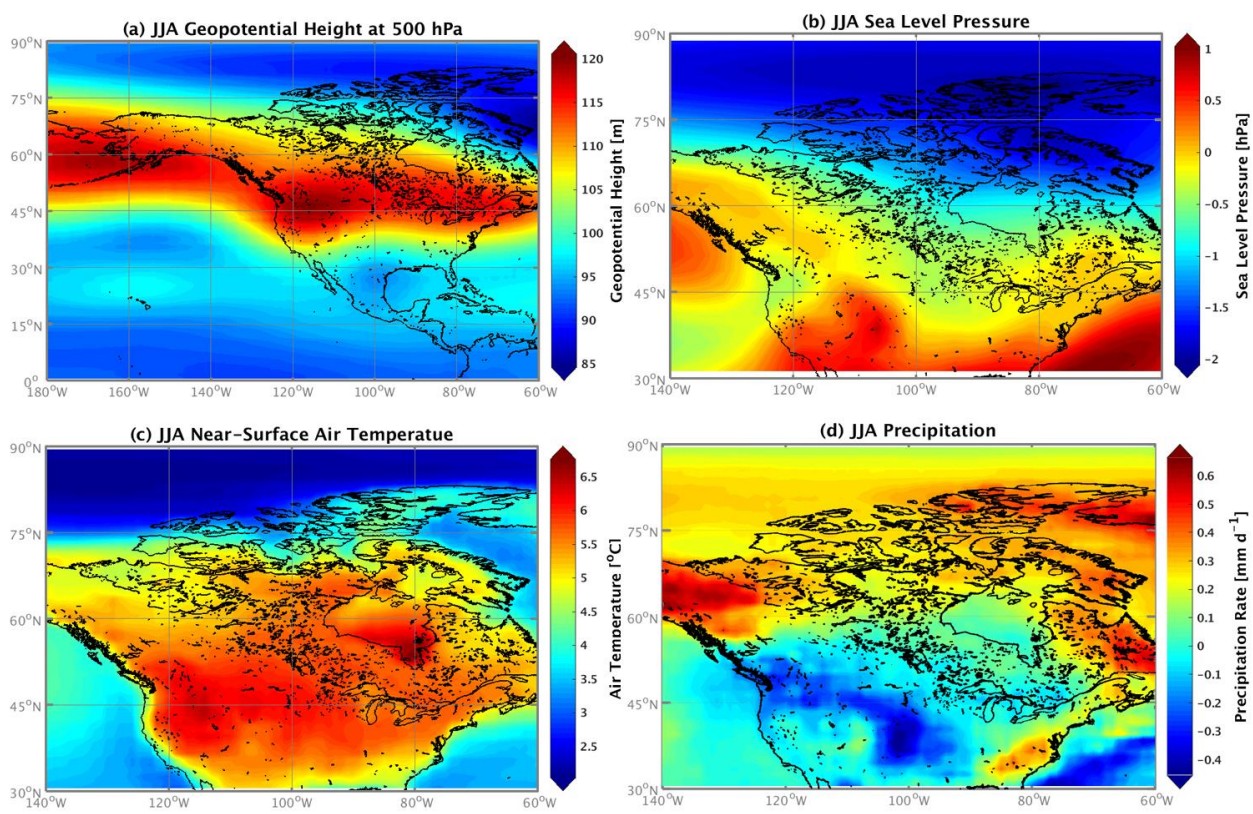

10    Figure 7: As in Fig. 2 but for summer (JJA).

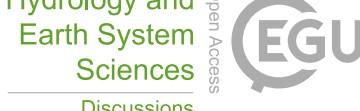



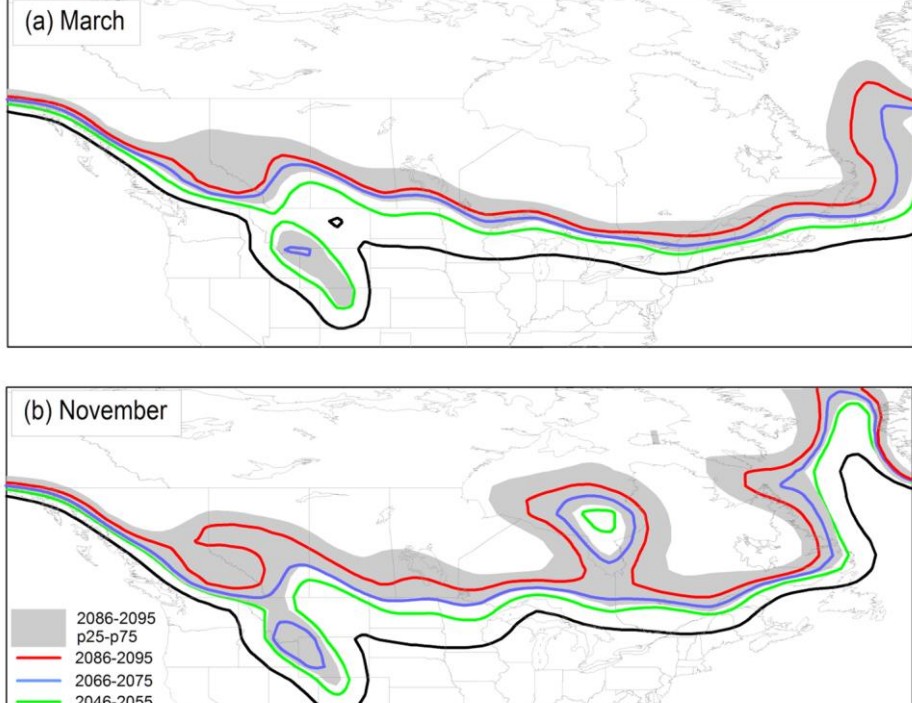

10  Figure 8: Locations of the 0°C isotherm during (a) March and (b) November over four different time periods. The
NCEP/NCAR re-analysis was used to compute locations for 1976-2005. The future time periods were computed by adding the
CMIP5 39-model ensemble median of the RCP8.5 10-y mean air temperature delta (future period – 1976-2005 historical)
projected for 3 different periods to the NCEP/NCAR climatology. The grey area shows the region bounded by the 0°C
isotherms in the 25$^{th}$ and 75$^{th}$ ensemble percentiles of the 2086-2095 mean air temperatures.





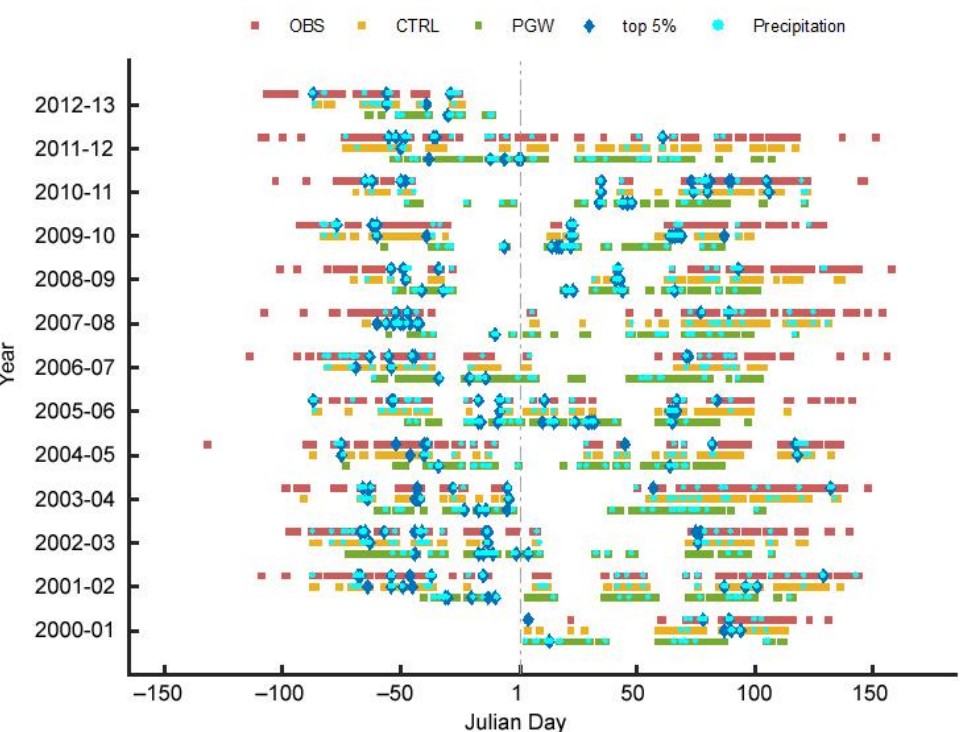

Figure 9: The occurrence of near 0°C conditions (temperatures in the range - 1°C ≤ T ≤ 1°C) by year and Julian day at Winnipeg

for the 2001-2012 period. Rows for each year indicate observations (OBS), WRF control (CTRL), and WRF PGW projections,

respectively. Dark blue dots indicate a near 0°C event within the top 5% in terms of duration and light blue symbols indicate

that precipitation occurred within an event. PGW projections are based on large scale temperatures and moisture conditions

15   occurring near the end of the century (Sect. 2).



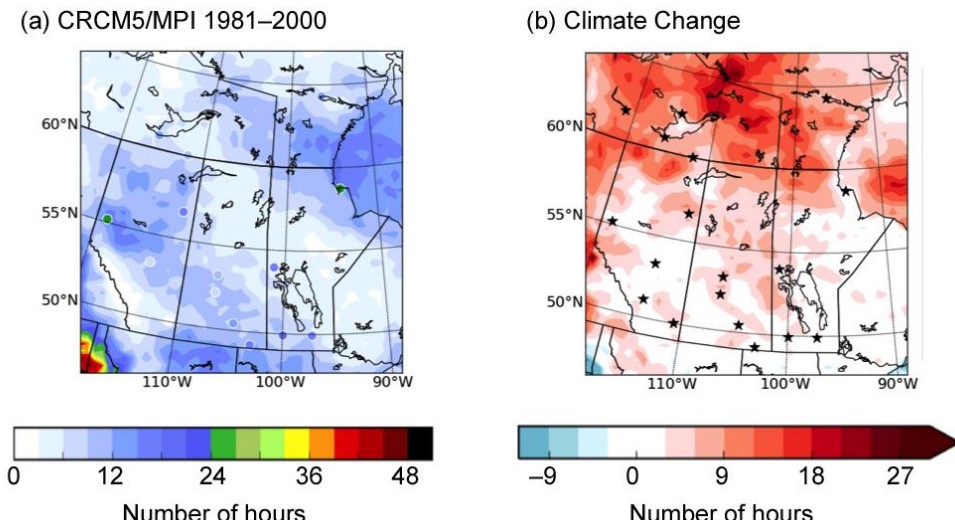

10  Figure 10: The mean annual number of hours of freezing rain (a) observed and simulated by the CRCM5 driven by MPI-ESM-
MR for the 1981-2000 period, (b) the change in the number of hours based on the difference between 2081-2100 and 1981-
2000 assuming the RCP8.5 scenario. Observational locations are indicated by circles in (a) and stars in (b).



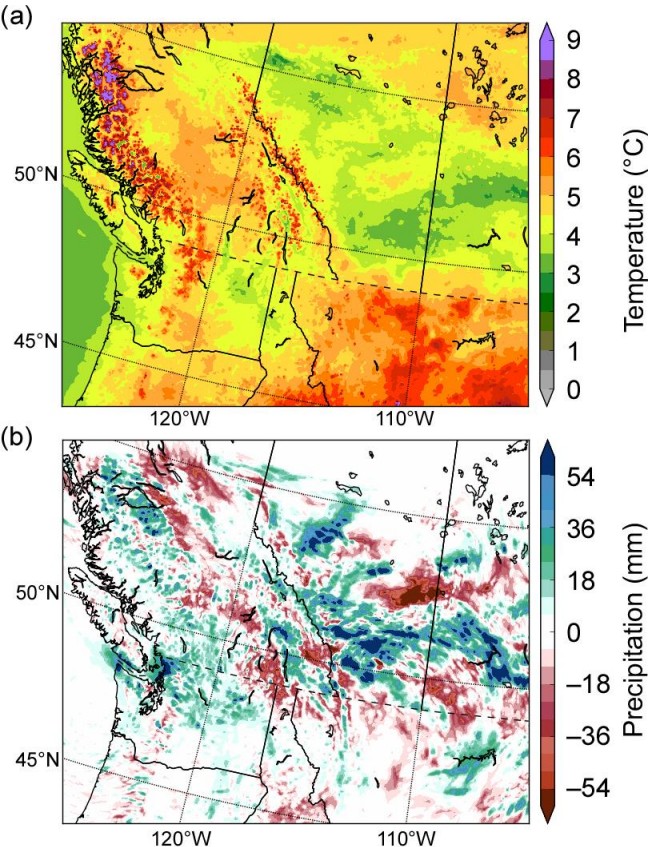

Figure 11: The June 19-22, 2013 Alberta flooding event comparing 4-day averaged (a) temperature (°C) and (b) precipitation

10   (mm) differences from the WRF control (CTRL) to the PGW simulations. PGW projections are based on large scale
temperatures and moisture conditions occurring near the end of the century (Sect. 2).




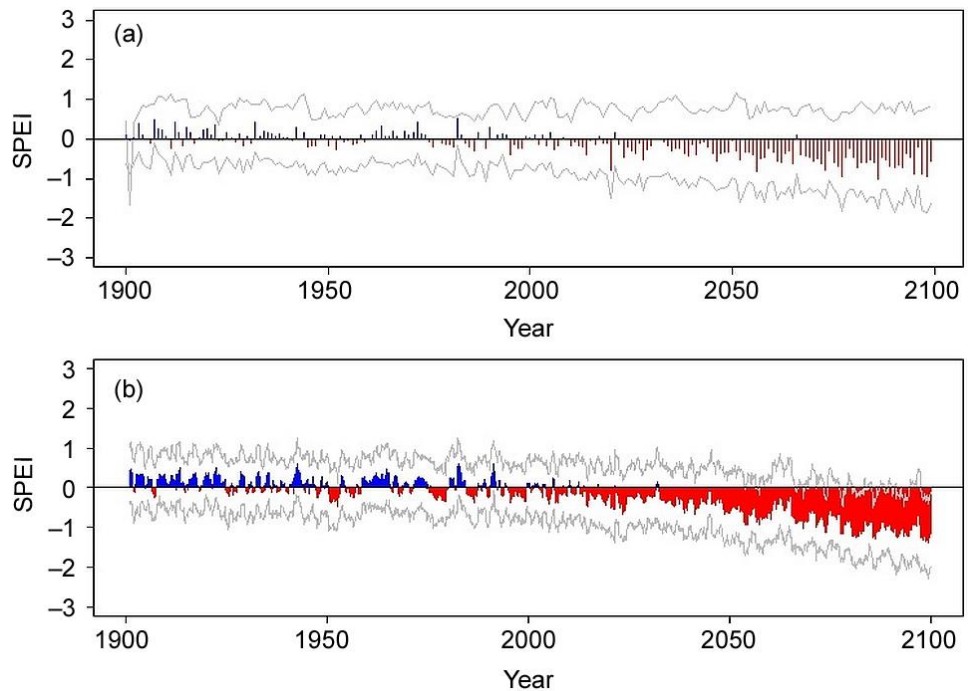

Figure 12: CMIP5 29-models ensemble medians of projected (a) summer (JJA) and (b) annual SPEI for RCP8.5 from 1900 to 2100 over the southern Prairies (defined in Sect. 2). Negative (red) values indicate surface water deficit relative to 1950-2005

10 conditions. Grey lines denote the 25th and 75th percentiles.



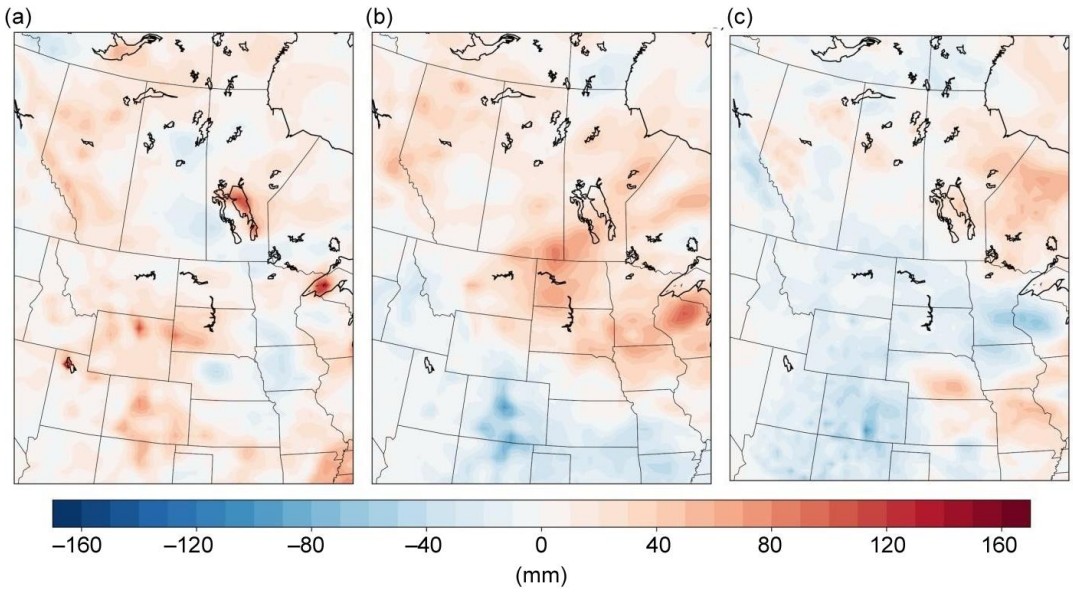

Figure 13: Changes in future warm season (JJA) total precipitation (mm) for three NARCCAP model pairs of (a) MM5-HadCM3, (b) MM5-CCSM and (c) HRM3-HadCM3. Positive values imply greater future precipitation (2041-2070 minus 1971-2000).



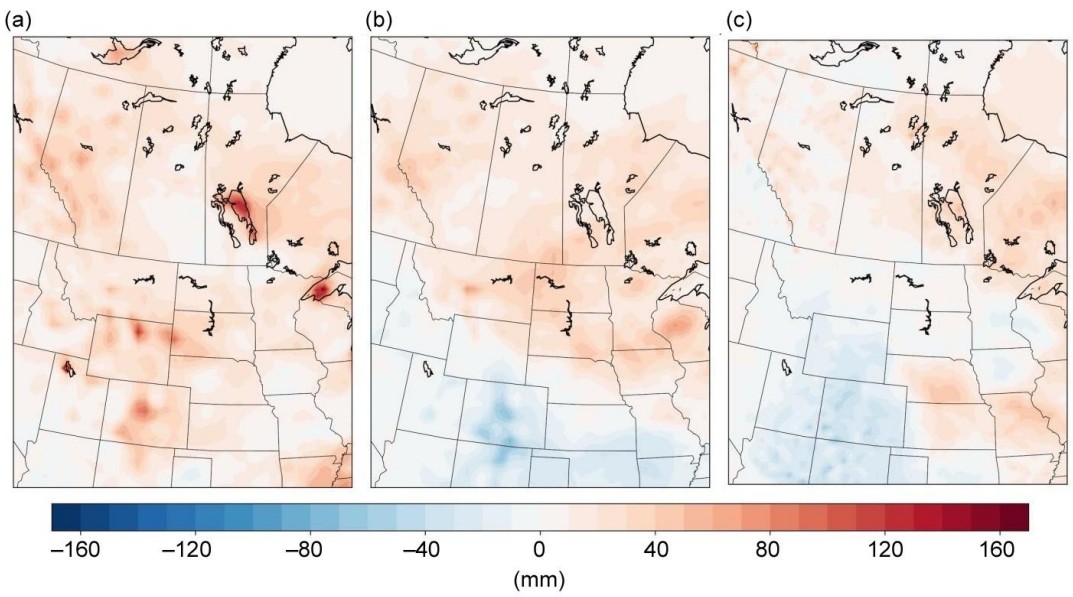

Figure 14: Changes in future warm season (JJA) convective precipitation (mm) for three NARCCAP model pairs of (a) MM5-
10  HadCM3, (b) MM5-CCSM and (c) HRM3-HadCM3. Positive values imply greater future precipitation (i.e. 2041-2070 minus
1971-2000).







Figure 15: Conceptual depiction of upper atmospheric, surface and phenomena changes projected under the RCP8.5 emissions scenario by the end of the century during (a) autumn (purple) and winter (blue) and (b) spring (green) and summer (brown). Upwards (downwards) pointed arrows indicate an expected increase (decrease). No arrow indicates no change and a question mark indicates uncertainty.