# Peer review of "A Review and Synthesis of Future Earth System Change in the Interior of Western Canada: Part I - Climate and Meteorology"

_Hydrology and Earth System Sciences, 2019_

## Referee Comment (RC1) · Anonymous Referee #1 · 10 Mar 2019

The manuscript "A Review and Synthesis of Future Earth System Change in the Interior of Western Canada: Part I - Climate and Meteorology" by Stewart et al. provides a review and shows novel research focused on climate change impacts in the Interior of Western Canada. The authors cover a large number of topics, which is necessary in a review article, but try to do this largely through their own research. This results in a fairly long article, which does not provide enough information to fully understand or correctly interpret the presented results. Also, the authors aim to connect results from single sections (e.g., changes in large-scale forcing and extreme events) without showing any physical evidence for such connection and often overinterpret results by making strong assumptions. I provide several suggestions on how to improve the manuscript in my points below. However, substantial effort is needed to improve the manuscript to a publishable level.

Major Comments:
1. The authors did not decide if they want to write a review or if they want to publish novel results. I am generally not against adding novel results to a review article but it makes a review paper much longer and reduces the accessibility of the presented information. The problem with adding your own results is that you have to clearly explain what you have done and you have to provide details on the experimental design. A good example is the use of the WRF-HRCONUS simulation. You are using this simulation in various places but the assumptions of the PGW method are only poorly described. A better strategy would be to refer to the existing publication that evaluated these simulations such as Prein et al. 2017, Dai et al. 2017, or Musselman et al. 2018.
   In general, making better use of existing literature and shortening the text to concisely review our state of knowledge, would make the article more easily accessible.
2. Related to the first point, the authors lose the reader due to the partly lengthy and unfocused assessments. You try to cover a wide range of topics with your own analysis and you are using a wide range of model experiments that have different strengths and weaknesses without addressing them adequately. Each of these topics could easily be its own manuscript. Relying more on existing analyses from e.g., the NARCCAM simulations instead of performing your own analysis would be beneficial. You barely mention any of the existing results from published NARCCAM papers in your review (e.g., Mearns et al. 2013, Gutowski et al. 2010, and others).
3. You mainly review changes in mean patterns and draw conclusions about their impacts on extremes. Extremes (e.g., flooding) are typically caused by weather patterns that are anomalous and changes in mean weather patterns might not be representative to derive estimates for extreme conditions. Please be careful with your interpretation of the effects of mean circulation pattern changes on extremes.
4. Similar to point 3, it seems like that many of your conclusions are speculative and are not based on tested causal relationships. I see no problem in discussing e.g., possible linkages between large-scale circulation changes on extreme phenomena but it should be clear that this is an untested hypothesis since you do not test these relationships in your manuscripts. Overall, I urge the authors to be more careful with the interpretation of their results.
5. You are using a large variety of modeling results with very different assumptions (e.g., CMIP5 vs. WRF-HRCONUS) and do not provide a critical review based on these assumptions.

E.g., In Section 3 you show significant changes in the large-scale circulation patterns and later you use the WRF-HRCONUS simulations, which assume no changes in circulation.

6. Why are you only using the RCP8.5 scenario? A review about climate change in a region should discuss the impacts of emission scenarios and internal climate variability on the presented results (e.g., Desser et al. 2012). Reading your article raises the impression that the presented changes are unavoidable and that emission scenario uncertainties are not important.

7. You misinterpret the rate of precipitation changes according to the Clausius-Clapeyron relationship. ~7 % precipitation increases per degree warming should be only realized for extreme precipitation whereas mean precipitation should theoretically increase by ~2 % per degree warming (e.g., Trenberth et al. 2003). The latter, however, is strongly regionally varying.

8. I encourage the authors to add a discussion on the state-of-the-art of climate change research in the CCRN region and to provide an assessment of future research needs. Highlighting future research needs could help to focus community research on the most important and least understood climate change processes in your region, which could be one of the main outcomes of this article.

Minor Comments:

P1-L26: "…projected to become stronger in each season…" what does that mean? In your assessments you show anomalies and for the reader it is hard to judge if the net effects increase or decrease synoptic scale forcing.

P1L-32: You have 7 references in your entire section 1. Many of the claims that you make here are without any reference.

P1L-32: I suggest to add a table that shows all of the here used model experiments.

P2-L3: "…will become wetter…" does this mean that the south is drying or that the north is becoming wetter regarding absolute precipitation?

P2-L5: "…likely have a huge impact" unless you cite references that show this I would suggest to use likely.

P2-L22-23: The temperature increase is likely a result of increased greenhouse gas forcing in your region and might be accelerated due the mentioned factors.

P3-L30: why did you use a 20/19 yearlong period instead of the standard 30-year long period? It seems like this makes the results harder to compare to the published literature.

P4-19-24: This simulation clearly needs more explanation about the PGW method and the included assumptions.

P4-L19: Why is there a C in Liu et al. (2016C)?

P5L13: Please explain the PNA index in the methods.

P5L18: "cooling is limited" where can I see this?

P5L22-23: 'and offsets its negative impacts' this needs some rewording.

P5L30: 'of historical molded? precipitation'

P6L1: and changes in the synoptic scale forcing

P6L6-7: I do not see any different trend in mid-century in Fig. 3b. The time series looks fairly linear to me.

P6L33: "could be able" would be more appropriate

P7L1: Please explain why you use a subset of 21 GCMs here.

P7L2: "In particular,…"

P7L14: Is this break in the time series really statistically significant?

P8L24-26: As above, is this change statistically significant. I have a hard time seeing it in Fig. 6b.

P9L14: I would add precipitation runoff to the list.

P9L20-23: Why did you not look at the northward shift in the zero-degree line in the models? Doing it the way you show it here might miss some important feedback mechanisms such as the snow-albedo feedback, which could accelerate the northward movement of the zero-degree isotherm.

P10L8: "in other" what? Areas?

P10L15: You introduced this simulation as WRF-HRCONUS on P4L24. Please be consistent with its naming.

P11L1-5: There is definitely a need for more explanation of these results. You spent page 5-9 of your manuscript talking about the importance of changes in large-scale drivers and then you present the results of the WRF-HRCONUS simulation, which assumes no changes in the large-scale drivers, without any discussion about this assumption.

P11L10: Why do you highlight the chinooks here while other processes might be equally important?

P11L30 to P12L5: I do not see the value of this regime classification compared to what you already said before in this section. This could be removed from the manuscript.

P12L31-32: I would be careful in interpreting these results. The precipitation patterns that you see can be caused by shifts in the most extreme storms, which cause result in areas with large increases and decreases next to each other. Shifts in the location of the precipitation maxima are typically chaotic and ensemble simulation would be necessary to identify the real climate change component of precipitation changes.

P13L10-13: This interpretation of the results is way to overconfident. First, how sure are you that your observed maxima is at exactly this location since you probably have a fairly sparse station network. Second, the effective resolution of climate models is typically >4-8 times its horizontal grid spacing. I would just state the model can capture the location of the precipitation maxima.

P14L12: How can I see that in Fig. 8a?

P14L19: what does 4.6/decade mean? Seasons per decade?

P14L20-21: Why are you comparing your results to the period 2081-2100? A 2.5-degree cooler FM period at the end of the century under RCP8.5 will be much warmer than a 2.5-degree cooler FM period in the past climate.

P14L27: Your summer cannot be typically linked to severe conditions. Severe conditions are per definition rare events and cannot be typical.

P16L7-9: This sounds like you are searching for an excuse to show mid-century results here. If it is that important to show mid- and end-century results you should also do this in the previous sections?

P16L19-30: Why are you discussing changes in mean summer precipitation again here? Some of the results look very different to the once you show in Fig. 10d. This makes it difficult for the reader to know, which interpretation they should belief in. I would suggest to remove this paragraph and maybe add some information to section 3.4.

P17L30-34: could be removed.

P18L1: This entire section has very little relevance to the CCRN region. Mentioning global average results on lightning can be easily misinterpreted as being relevant for Canada, which they most like are not. This section can be easily condensed to a single paragraph with the main message that we simply do not know a lot about future lightning in Canada.

P19L18: How confident are you here based on the large uncertainties in future lighting projections?

P19L20: What about the future availability of fuel?

P20L8: You could mention Fig. 15a here.

P20L12: Fig. 15b

P20L20: I think "link" is too strong here. You did not show any causal relationships in your analysis and your interpretation are mostly based on your interpretation of the results.

P20L23: Understanding the origin of changes would be very important as well.

P21L3: "...natural and anthropogenic factors."

P21L11: please replace "dramatic" with something more quantifiable.

P21L11-12: Which 4 models are you talking about?

Figures:

Fig. 2,4,5,7: The rainbow colortable does not allow to see a lot of details (e.g., Fig. 2 d shows a red area along the west coast but changes in the focus region are shown in a single blue tone. Please select a more appropriate colortable and use fixed color bar ranges in all 4 figures. At the moment it is very hard to compare them. Also, adding a significant layer on top of it (similar to what the IPCC uses in their assessment) would be highly beneficial.

Fig.10: Please adjust the color range. E.g., Fig. 10a has only one red spot in the lower left corner. Gradient would be much easier to see if you would reduce the maximum from 48 to 24 hours.

Fig. 13-14: Please reduce your color range also in these figures.

Fig. 15: This figure could be better imbedded in your manuscript (you only mention it once).

Literature:

Deser, C., Phillips, A., Bourdette, V. and Teng, H., 2012. Uncertainty in climate change projections: the role of internal variability. Climate dynamics, 38(3-4), pp.527-546.

Trenberth, K.E., Dai, A., Rasmussen, R.M. and Parsons, D.B., 2003. The changing character of precipitation. Bulletin of the American Meteorological Society, 84(9), pp.1205-1218.

Prein, A.F., Rasmussen, R.M., Ikeda, K., Liu, C., Clark, M.P. and Holland, G.J., 2017. The future intensification of hourly precipitation extremes. Nature Climate Change, 7(1), p.48.

Musselman, K.N., Lehner, F., Ikeda, K., Clark, M.P., Prein, A.F., Liu, C., Barlage, M. and Rasmussen, R., 2018. Projected increases and shifts in rain-on-snow flood risk over western North America. Nature Climate Change, 8(9), p.808.

Dai, A., Rasmussen, R.M., Liu, C., Ikeda, K. and Prein, A.F., 2017. A new mechanism for warm-season precipitation response to global warming based on convection-permitting simulations. Climate Dynamics, pp.1-26.

Mearns, L.O., Sain, S., Leung, L.R., Bukovsky, M.S., McGinnis, S., Biner, S., Caya, D., Arritt, R.W., Gutowski, W., Takle, E. and Snyder, M., 2013. Climate change projections of the North American regional climate change assessment program (NARCCAP). Climatic Change, 120(4), pp.965-975.

Gutowski Jr, W.J., Arritt, R.W., Kawazoe, S., Flory, D.M., Takle, E.S., Biner, S., Caya, D., Jones, R.G., Laprise, R., Leung, L.R. and Mearns, L.O., 2010. Regional extreme monthly precipitation simulated by NARCCAP RCMs. Journal of Hydrometeorology, 11(6), pp.1373-1379.

---

## Referee Comment (RC2) · Anonymous Referee #2 · 21 Mar 2019

This review is written by researchers with ample experience in the Canadian climate. The manuscript aims at synthesizing expected changes in future regional climate of western Canada due to anthropogenic influences. The authors provide a wealth of information selecting specific processes or phenomena of relevance to the region. In doing so, they define the basis for research priorities to advance on the knowledge of the regional impacts of a changing climate. I found the review and synthesis to be useful, exhaustive and well documented.

As with many review articles written by several authors, there tends to be some inconsistency among the different sections, with some easier to read than others. For

example, the motivation section is clear, defined and even entertaining. On the other hand, section 3 could benefit from refining the main concepts and doing a better link between the discussion and the supporting figures. The manuscript has value and quality. It will be even more attractive once the sections that need improvement are refined. I recommend that the manuscript is approved after the following points are addressed.

1. Section 3 seems to lack an introduction and jumps directly to describe changes in large-scale seasonal patterns. As the discussion is based on changes or anomalies, it would be useful to start with a description of the present climate and more specifically of the PNA pattern as known now. This section also assumes that many features are known to most readers. The second paragraph in page 5 is an example of frequent statements presented without a clear argument: "Projected regional climate responses to the circulation changes are consistent with those found during negative PNA, but shifted in association with the projected circulation features." Not everybody knows the regional features of the negative PNA, or how they would be shifted due to changes in the projections. Is there a way to infer the changes in cold advection from the figures?

Unfortunately, this is not a matter rewriting a couple of paragraphs. Rather, it is about how the (quite complex) concepts of the PNA pattern and its future changes are presented for the four seasons. My suggestion is that the authors simplify the text by limiting the discussion to key issues that can be easily linked to the figures or adequate references.

I suggest following a similar approach to that in section 4.1.1. The discussion of changes in the 0 C isotherm is straightforward and supported by a figure that is easy to follow (Fig. 8).

2. I have difficulty agreeing with the interpretation of Figure 13. "Consistent" features are described for relatively small regions of the domain (e.g., central and northern Manitoba ot north central Alberta), but the main issue that seems to be ignored is that

there are important inconsistencies over large areas of the domain. These large-scale differences among the models can suggest that agreements over the small regions are just the result of chance. An objective approach is needed to separate the wheat from the chaff.

3. It is discussed that given the lack of lightning data, a proxy based on cloud-top heights has been used by Price and Rind (1994). Has this approach been validated in any manner?

---

## Author Comment (AC1) · 20 Jun 2019

**REVIEWERS 1 AND 2: COMMENTS AND RESPONSES**

**REVIEWER 1**

The manuscript "A Review and Synthesis of Future Earth System Change in the Interior of Western Canada: Part I - Climate and Meteorology" by Stewart et al. provides a review and shows novel research focused on climate change impacts in the Interior of Western Canada. The authors cover a large number of topics, which is necessary in a review article, but try to do this largely through their own research. This results in a fairly long article, which does not provide enough information to fully understand or correctly interpret the presented results. Also, the authors aim to connect results from single sections (e.g., changes in large-scale forcing and extreme events) without showing any physical evidence for such connection and often over interpret results by making strong assumptions. I provide several suggestions on how to improve the manuscript in my points below. However, substantial effort is needed to improve the manuscript to a publishable level.

Major Comments:

1. The authors did not decide if they want to write a review or if they want to publish novel results. I am generally not against adding novel results to a review article but it makes a review paper much longer and reduces the accessibility of the presented information. The problem with adding your own results is that you have to clearly explain what you have done and you have to provide details on the experimental design. A good example is the use of the WRF-HRCONUS simulation. You are using this simulation in various places but the assumptions of the PGW method are only poorly described. A better strategy would be to refer to the existing publication that evaluated these simulations such as Prein et al. 2017, Dai et al. 2017, or Musselman et al. 2018.

In general, making better use of existing literature and shortening the text to concisely review our state of knowledge, would make the article more easily accessible.

**Response: We did update the title of the article to better reflect our scope which is largely based on research related to the Changing Cold Regions Network (CCRN). Additional references have been added as suggested, including a new ECCC climate change summary report, and the text has been improved in line with these suggestions. To narrow the article's focus, we have taken out all WRF-CONUS information.**

**New and/or Revised Text: The title and the following paragraphs have been revised or are entirely new to give the reader a better background for this article.**
**Title: "Summary and Synthesis of Changing Cold Regions Network (CCRN) Research in the Interior of Western Canada: Part 1 - Projected Climate and Meteorology"**

**New/updated text: "Climate and its changes are having huge impacts everywhere. A particular 'hotspot' in Canada in terms of recent temperature changes and projections of continuation is the central part of western Canada and its extension to the Arctic Ocean (DeBeer et al., 2016). Although there is widespread consensus that warming will continue,**

**there is considerable uncertainty in its magnitude and distribution in time and space. There is even greater uncertainty in terms of precipitation although it is very likely that there will be less snow and more rain, and the north will become wetter (Bush and Lemmen, 2019).**

**All of these changes will likely have a huge impact on water resources, cryosphere and ecosystems. In terms of hydrology, this includes the amount of water as well as the timing of its peak flow; in terms of the cryosphere, this includes the fate of numerous glaciers, regions of permafrost and the duration and amount of snow; in terms of ecosystems, this includes movement of grasslands, tundra, shrubs and boreal forests (Bush and Lemmen, 2019).**

**These critical issues have been the motivation for substantial climate-related research within the central part of western Canada. Much of this was organized within collaborative multi-year projects. The first was the Mackenzie GEWEX Study (MAGS), under the auspices of the Global Energy and Water Exchanges (GEWEX) project of the World Climate Research Programme, that brought together atmospheric and hydrological researchers to examine the cycling of water within the Mackenzie River basin (Stewart et al., 1998; Woo et al., 2008). This was followed by, for example, the Drought Research Initiative (DRI) that examined atmospheric, hydrologic and land surface processes associated with a devastating 1999-2005 drought across the Canadian Prairies (Stewart et al., 2011; Hanesiak et al., 2011). In parallel, the Western Canadian Cryospheric Network (WC2N; http://wc2n.unbc.ca) and the Improved Processes and Parameterization for Prediction in Cold Regions Hydrology Network (IP3; www.usask.ca/ip3) examined hydrologic and cryospheric issues affecting the western Canadian Cordillera. Major scientific progress was made within these projects, as largely summarized in DeBeer et al. (2016), but their main focus was examining the past and present climate and improving the understanding and modelling of key processes with relatively little focus on future conditions."**

2. Related to the first point, the authors lose the reader due to the partly lengthy and unfocused assessments. You try to cover a wide range of topics with your own analysis and you are using a wide range of model experiments that have different strengths and weaknesses without addressing them adequately. Each of these topics could easily be its own manuscript. Relying more on existing analyses from e.g., the NARCCAM simulations instead of performing your own analysis would be beneficial. You barely mention any of the existing results from published NARCCAM papers in your review (e.g., Mearns et al. 2013, Gutowski et al. 2010, and others).

**Response: We have updated the entire convection and hail section (4.3.2) and have added information about existing NARCCAP articles in comparison to our results as suggested by the reviewer, where possible. Many existing articles focus on different seasons or span the entire annual period, so direct comparisons are not possible, except with the Mearns et al. (2013) article and part of the Mailhot et al. (2011) article. Our analysis and review is novel as no other studies have focused purely on convective precipitation and hail over the**

**Canadian Prairies. We have also completely removed the total precipitation analysis since it is largely redundant to Mearns et al. (2013).**

**New and Revised Text: "NARCCAP historic (1971 to 2000) and mid-century future (2041 to 2070) model output was used to assess future changes in convective precipitation and hail over the Canadian Prairies, southern Northwest Territories and U.S. northern plains. Convective precipitation is defined to occur when the model convective scheme is triggered to release latent energy and convective instability through simulated vertical motion. Brimelow et al. (2017) suggested that the three most consistent NARCCAP model pairings to assess convective precipitation and hail for the regions of interest herein, included MM5-HadCM3, MM5-CCSM and HRM3-HadCM3, based on their ability to reproduce the precipitation climatology. No other NARCCAP studies focused on hail or warm season convection-only precipitation, although Mearns et al. (2013) and Mailhot et al. (2011) looked at ensemble summer total precipitation and annual maximum precipitation, respectively, while other studies focused on other seasons (e.g. Gutowski et al., 2010; Kawazoe and Gutowski, 2013).**

**Changes in future summer (JJA) convective precipitation are shown in Fig. 13 for three NARCCAP model pairs under the SRES A2 scenario. All three model pairs show increases over much of the Prairies but with varying amounts (near zero to 50 mm). MM5-CCSM and HRM3-HadCM3 are consistent with CMIP5 RCP8.5 results for the same future period (not shown) and the spatial patterns in other studies (i.e. increases in Canada but decreases in central/southern U.S. Plains) (e.g. Mearns et al., 2013; Mailhot et al., 2011). These results are also consistent with general increases in CAPE and surface dew points in a warming climate over much of the Prairies (e.g. Brimelow et al., 2017)."**

3. You mainly review changes in mean patterns and draw conclusions about their impacts on extremes. Extremes (e.g., flooding) are typically caused by weather patterns that are anomalous and changes in mean weather patterns might not be representative to derive estimates for extreme conditions. Please be careful with your interpretation of the effects of mean circulation pattern changes on extremes.

**Response: Good Point. We were implicitly assuming that significant future changes in the mean circulation patterns are, at least partially, manifested in changes of the intensity and/or the frequency of similar anomalous circulation patterns with respect to current conditions. Such assumptions are now explicitly stated in the revision. We also added new analysis and discussions to verify these assumptions.**

**New and/or Revised Text: In section 3: "Results in Szeto (2008) and Szeto et al. (2015, 2016) show that significant correlations are exhibited between hydroclimate variables in the domain and the intensities of these seasonal circulation features. In addition, cold- and warm-season extreme conditions are often associated with intense respective circulation anomalies as reflected in extremity of the corresponding PNA and H-indices that measure the intensity of these circulation anomalies. Since there are physical bases for such associations between regional climate variability and extremes in the domain and these large-scale circulation anomalies, it is not unreasonable to assume that such relationships**

will also hold for future changes in these large-scale drivers and the regional climate responses. We further assume that any significant future changes in the mean circulation pattern will, at least partially, manifested in changes of the intensity and/or the frequency of similar anomalous circulation patterns with respect to current conditions. This assumption will be verified with model data in the following and the validity of this assumption will lend support to the idea that changes in the mean circulation could be linked to extreme climate responses in the area."

New and/or Revised Text related to the validation of the assumption:

Autumn: "The mean near-neutral PNA condition that characterizes the first half of the century is replaced by mean negative (-0.26) conditions and 9 out of the 10 strongest negative PNA autumns are found after 2060; this latter result provides support to the assumption that changes in the mean circulation pattern is partly manifested in changes in the frequency and/or intensity of the extreme similar anomaly patterns. "

Winter: "An increasing trend, albeit merely significant at the 10% level, in the ensemble-median DJF PNA index is found only during the first half of the century (Fig. 3b). An abrupt 'jump' is projected to occur at the end of the increasing trend during the late 2050s where the piecewise linear regression lines over the two periods is separated by a statistically significant gap that is larger than the variability of the index. The significance of the mid-century change is reflected in the marked increase of mean PNA index from 0.10 in the first half of the century to 0.37 in the second half as well as in the fact that all of the 10 projected strongest PNA winters occur after 2060. This latter result also provides support to the assumption we made on the relationship between changes in the mean and extreme patterns."

Spring: "The mean magnitude of the low (i.e., negative H-index) is projected to intensify mainly during the mid-century (Fig. 6a). In fact, the 20-y mean H (not shown) decreases by 15 m (from -10 m to -25 m) during 2045-2065, suggesting that some radical changes in the MAM large-scale circulations are projected to occur mid-century. It is noteworthy that this mid-century decrease in 20-y mean H is even larger than the standard deviation of H over the century (13.9 m). In addition, 8 of the 10 lowest H-index springs, i.e., springs that are likely to be associated with extreme wet conditions over the Prairies, occur after 2040."

Summer: It is also notable that all of the top 10 highest H-index summers, i.e., summers with extreme warmth and dryness likely occurring in the Prairies, occur after 2050."

4. Similar to point 3, it seems like that many of your conclusions are speculative and are not based on tested causal relationships. I see no problem in discussing e.g., possible linkages between large-scale circulation changes on extreme phenomena but it should be clear that this is an untested hypothesis since you do not test these relationships in your manuscripts. Overall, I urge the authors to be more careful with the interpretation of their results.

Response: Some of the linkages between changes in the large-scale patterns and regional extremes are based on correlations or composite analyses of historical hydroclimate events

**in the regions that are presented in the references (e.g. Shabbar et al., 2011; Szeto et al., 2015, 2016). We agree that we have not tested if these relationships will also hold under climate change in the manuscript. We had addressed these points in our Concluding Remarks but we appreciate the requirement to improve this early on as well as to improve the text in the final section.**

**We have added more information on this aspect in the revision.**

**New and/or Revised Text: New text in Section 3: "Results in Szeto (2008) and Szeto et al. (2015, 2016) show that significant correlations are exhibited between hydroclimate variables in the domain and the intensities of these seasonal circulation features. In addition, cold- and warm-season extreme conditions are often associated with intense respective circulation anomalies as reflected in extremity of the corresponding PNA and H-indices that measure the intensity of these circulation anomalies. Since there are physical bases for such associations between regional climate variability and extremes in the domain and these large-scale circulation anomalies, it is not unreasonable to assume that such relationships will also hold for future changes in these large-scale drivers and the regional climate responses. We further assume that any significant future changes in the mean circulation pattern will, at least partially, be manifested in changes of the intensity and/or the frequency of similar anomalous circulation patterns with respect to current conditions. This assumption will be verified with model data in the following and the validity of this assumption will lend support to the idea that changes in the mean circulation could be linked to extreme climate responses in the area."**

**Revised text in Concluding Remarks: "The further characterization and determination of origins of large-scale circulation changes and whether the associations between these large-scale drivers and smaller scale phenomena that were established basing on historical data would change in the future need to be investigated to a greater extent."**

5. You are using a large variety of modeling results with very different assumptions (e.g., CMIP5 vs. WRF-HRCONUS) and do not provide a critical review based on these assumptions. E.g., In Section 3 you show significant changes in the large-scale circulation patterns and later you use the WRF-HRCONUS simulations, which assume no changes in circulation.

**Response: To narrow the article's focus, WRF-HRCONUS information has been removed.**

6. Why are you only using the RCP8.5 scenario? A review about climate change in a region should discuss the impacts of emission scenarios and internal climate variability on the presented results (e.g., Desser et al. 2012). Reading your article raises the impression that the presented changes are unavoidable and that emission scenario uncertainties are not important.

**Response: We revised our text to address this issue. New text has been added to Section 2 and Concluding Remarks text has been revised to more clearly address this issue.**

**New and/or Revised Text: In Section 2: "This approach does not capture the impacts arising from a full range of emission scenarios. It nonetheless allows for a physically-based analysis and interpretation of a business-as usual scenario although it is recognized that there is considerable uncertainty within this one scenario. The results can be used as a basis for follow-on studies that explore a wider range of possible futures."**

**This had already been mentioned in Concluding Remarks; this wording was also enhanced. "In particular, there is a range of possible scenarios and, within each, there are numerous model products. Large scale circulation changes in this article were largely addressed using ensemble information; different emissions scenario RCP8.5 contributing models may have developed somewhat different patterns. The article furthermore was based on the interpretation of available information; confirmation of relations between, for example, large scale circulations and smaller scale phenomena is needed."**

7. You misinterpret the rate of precipitation changes according to the Clausius-Clapeyron relationship. ~7 % precipitation increases per degree warming should be only realized for extreme precipitation whereas mean precipitation should theoretically increase by ~2 % per degree warming (e.g., Trenberth et al. 2003). The latter, however, is strongly regionally varying.

**Response: Agreed. The text related to this point has been removed.**

8. I encourage the authors to add a discussion on the state-of-the-art of climate change research in the CCRN region and to provide an assessment of future research needs. Highlighting future research needs could help to focus community research on the most important and least understood climate change processes in your region, which could be one of the main outcomes of this article.

**Response: More background on previous research within the CCRN region has been added. In addition, a new ECCC climate change summary report has just been released and it provides a wealth of relevant material and is now cited. Further insight on needed future research has been more added beyond what was mentioned previously within Concluding Remarks.**

**New and/or Revised Text: The revised text with more background on CCRN-related research was covered under Major Comment 1 (shown above).**

**In Concluding Remarks, revised and additional text is as follows: "This study has examined conditions mainly applicable towards the end of the century over the CCRN domain, largely using one business-as-usual emissions scenario (RCP8.5), and placed these within a strong physical basis. Although not as extensive, some attention was paid to the evolution of these conditions. These analyses led to the development of a physically-based conceptual framework relating large scale atmospheric change to smaller scale associated features. Because of projected seasonal shifts in circulations and temperature, four conceptual depictions were developed to account for changes in associated phenomena.**

**Although these syntheses are based on solid physical interpretation, they have limitations. First, the inter-scenario and inter-model variability of both the large-scale drivers and regional responses need to be better assessed. The article furthermore was largely based on the interpretation of available information on the intra-annual time scale and Pacific North-America domain; The analysis of this information mainly focused on their multi-year smoothed characteristics and some rudimentary analyses of the evolving pattern of large scale upper atmospheric and surface drivers. The further characterization and determination of origins of large-scale circulation changes and the assessment of whether associations between these large-scale drivers and smaller scale phenomena that were established using historical data would change in the future need to be investigated to a greater extent. Preliminary results revealed many 'surprises' which include radically different seasonal regional responses to flips in circulation patterns with season. In addition, statistically significant differences identified in the trends and other statistics of upper circulation patterns before and after mid-century suggest possible regime shifts of the seasonal large-scale drivers during this timeframe. Further research to elucidate the nature of such abrupt changes and to examine how such nonlinear large-scale responses to climate change are simulated in different models is critical for future improvements of climate change projections.**

**In addition, the analyses mainly relied on coarse resolution model outputs and future studies need to address critical issues in more detail. In particular, CMIP5 models may not properly account for all critical processes in the atmosphere, surface and boundary layer; their projections may lead to different hydroclimatic conditions than those from finer-resolution regional models. Higher resolution model projections are of particular importance for the region because many of the hydroclimate extremes in the area are related to frontal and organized convective systems that develop over the complex terrains which characterize the region. Moreover, numerous feedbacks from the evolving land surface, including vegetation changes, snowcover and freeze-thaw processes, need to be better accounted for; these affect atmospheric circulations, storms and precipitation distributions. Other surface-related feedbacks involve shifting oceanic circulations and sea ice evolution. The analyses furthermore did not directly consider the critical role of clouds that has been shown in governing the atmospheric and surface water and energy budgets of the region; this certainly needs to be addressed. A related issue is ensuring that vertical atmospheric profiles are well handled over this evolving cold climate region; this is critical for atmospheric stability considerations which influence many atmospheric phenomena including precipitation distributions. Progress made here is therefore an important accomplishment that future studies can build on."**

Minor Comments:

**P1-L26**: "…projected to become stronger in each season…" what does that mean? In your assessments you show anomalies and for the reader it is hard to judge if the net effects increase or decrease synoptic scale forcing.

**Response: That was incorrectly stated and the text has been revised.**

**New and/or Revised Text: "Large scale atmospheric circulations affecting this region are projected to shift in each season…"**

*P1L-32*: You have 7 references in your entire section 1. Many of the claims that you make here are without any reference.

**Response: A new ECCC climate change summary report is now cited and this has brought together many articles. Additional references have also been added.**

**New and/or Revised Text: This new ECCC report is Bush and Lemmen (2019) and it, as well as other references, have been inserted in several spots with examples shown in response to Reviewer Major Comment 1 above.**

*P1L-32:* I suggest to add a table that shows all of the here used model experiments.

**Response: Good suggestion.**

**New and/or Revised Text: The new table is shown below.**

**Table 1: Model products used in this study as well as time periods used for mean historical and future conditions. Acronyms are defined in the text.**

| Model | Scenario | Time Periods | |
|---|---|---|---|
| CMIP 5 | RCP8.5 | 1981-2000 | 2081-2099 |
| CRCM5 | RCP8.5 | 1981-2000 | 2081-2100 |
| NARCCAP | SRES A2 | 1971-2000 | 2041-2070 |
| NCEP/NCAR | - | 1976-2005 | - |

*P2-L3:* "…will become wetter…" does this mean that the south is drying or that the north is becoming wetter regarding absolute precipitation?

**Response: We shortened the sentence to just refer to the north becoming wetter.**

**New and/or Revised Text: "...the north will become wetter (Bush and Lemmen, 2019)."**

*P2-L5:* "…likely have a huge impact" unless you cite references that show this I would suggest to use likely.

**Response: Good point. The word 'likely' has been included.**

**New and/or Revised Text: "All of these changes will likely have a huge impact…"**

*P2-L22-23*: The temperature increase is likely a result of increased greenhouse gas forcing in your region and might be accelerated due the mentioned factors.

**Response: We agree that the temperature increase is likely a result of increased greenhouse gases. We added a reference for this and we added an additional comment regarding ice albedo feedback.**

**New and/or Revised Text: "These temperature increases, believed to be mainly due to increased greenhouse and associated atmospheric factors (Bush and Lemmen, 2019), have been associated with changes to precipitation regimes and unambiguous declines in snow cover depth, persistence, and spatial extent and it has caused mountain glaciers to recede at all latitudes, permafrost to thaw at its southern limit, and active layers over permafrost to thicken. Some of these many changes might have accelerated temperature increases largely through ice albedo feedbacks."**

_P3-L30:_ Why did you use a 20/19 year long period instead of the standard 30-year long period? It seems like this makes the results harder to compare to the published literature.

**Response: The use of a 20/19 year long period is not unusual. For example, an assessment entitled Canada's Changing Climate Report has just been released by Environment and Climate Change Canada (Bush and Lemmen, 2019). It also utilized 20 year periods and this has now been noted.**

**New and/or Revised Text: "Similar 20 year long periods were used within the recent Canada's Changing Climate Report (Bush and Lemmen, 2019)."**

_P4-19-24_: This simulation clearly needs more explanation about the PGW method and the included assumptions.

**Response: WRF-HRCONUS information has been removed.**

_P4-L19_: Why is there a C in Liu et al. (2016C)?

**Response: There were two articles published in 2016 with two different lead authors with last name LIU. The first names of these authors begin with A and C. This is no longer an issue since all WRF-HRCONUS information has been removed.**

_P5L13:_ Please explain the PNA index in the methods.

**Response: Good suggestion. Brief physical interpretations of the PNA and H-Index have been added in Section 2.**

**New and/or Revised Text: "In particular, the projected changes of these cold and warm season circulation features are examined by calculating the "4 point" PNA index as formulated in Wallace and Gutzler (1981) and the H-index introduced in Szeto et al. (2016) using the CMIP5 500 hPa geopotential height data, respectively. The "4-point" PNA index quantifies the amplitude of the PNA wave train by comparing the 500 hPa height at four different fixed locations and the H-index quantifies the magnitude of an upper-level**

circulation feature by comparing the height field at the center and enclosing areas of the feature."

*P5L18:* "cooling is limited" where can I see this?

**Response: This phrase has been deleted from the text.**

*P5L22-23*: 'and offsets its negative impacts' this needs some rewording.

**Response: This wording has been changed to "...and partly offsets its detrimental effects on the low-level background baroclinicity..."**

**New and/or Revised Text: The warming over the south effectively reduces the S-N gradient of net anthropogenic warming (Fig. 2c) and partly offsets its detrimental effects on the low-level background baroclinicity and synoptic storms that affect southwestern Canada.**

*P5L30:* 'of historical modeled? precipitation'

**Response: Good suggestion. This clarifying phrase was added.**

**New and/or Revised Text: "... the increase is larger than the natural variability of historical modeled precipitation for the region…"**

*P6L1:* and changes in the synoptic scale forcing

**Response: Good point. This wording has been inserted into the text as suggested.**

**New and/or Revised Text: "...complex topography of the region, as well as changes in the synoptic scale forcing…"**

*P6L6-7:* I do not see any different trend in mid-century in Fig. 3b. The time series looks fairly linear to me.

**Response: We have added new piecewise regression analyses for the time series in the revision and both Figs. 3a,b and 6a,b have been revised to include these new results. Changes in the trends are more apparent in the new figures and are discussed in the revised text.**

**New and Revised Text: The new Figures 3a,b and 6a,b are shown below (in order).**

[revised manuscript text omitted]

_P6L33:_ "could be able" would be more appropriate

**Response: Good point. Done.**

**New and/or Revised Text: "...moisture-laden southern systems could be able to track…"**

_P7L1:_ Please explain why you use a subset of 21 GCMs here.

**Response: We only have daily data for 21 models on our system for this analysis and we did not have the resources to acquire the additional data for the rest of the models. That is why we had referred to this analysis as being "preliminary". An additional sentence has been added.**

**New and/or Revised Text: "It is a preliminary analysis because we only have access to daily data for 21 out of the 39 models for this analysis."**

*P7L2:* "In particular…"

**Response: We have corrected that typo. The word 'in' appears in our original text but the first letter (i) was somehow missing in the submitted version. Strange.**

**New and/or Revised Text: "In particular…"**

*P7L14:* Is this break in the time series really statistically significant?

**Response: Please see reply to *P6L6-7.***

*P8L24-26:* As above, is this change statistically significant. I have a hard time seeing it in Fig. 6b.

**Response: Please see response to *P6L6-7*.**

*P9L14:* I would add precipitation runoff to the list.

**Response: Good suggestion. Done.**

**New and/or Revised Text: "The location of the 0°C isotherm is a critical aspect of this region's climate. It is closely linked with the melting of snow at the surface which in turn affects albedo, land-atmospheric energy exchange and precipitation runoff…"**

*P9L20-23:* Why did you not look at the northward shift in the zero-degree line in the models? Doing it the way you show it here might miss some important feedback mechanisms such as the snow-albedo feedback, which could accelerate the northward movement of the zero degree isotherm.

**Response: We did use model products to determine the northward movement. We are not sure what is meant by the question.**

*P10L8:* "in other" what? Areas?

**Response: We mean other events in the same area. This has been clarified.**

**New and/or Revised Text: "...in the spring of 2015 in the Kananaskis area of the Alberta foothills, a mixture of rain and snow has been observed at temperatures as high as 9°C in some events, whereas it only occurred below 2-3°C in other events…"**

*P10L15:* You introduced this simulation as WRF-HRCONUS on P4L24. Please be consistent with its naming.

**Response: All WRF-HRCONUS information has been removed.**

***P11L1-5:*** There is definitely a need for more explanation of these results. You spent page 5-9 of your manuscript talking about the importance of changes in large-scale drivers and then you present the results of the WRF-HRCONUS simulation, which assumes no changes in the large scale drivers, without any discussion about this assumption.

**Response: All WRF HRCONUS information has been removed.**

***P11L10:*** Why do you highlight the chinooks here while other processes might be equally important?

**Response: We now point out that extra-tropical cyclones as well as chinooks are critical.**

**New and/or Revised Text: "... typically include extra-tropical cyclones with warm fronts although chinook-associated patterns are also important…"**

***P11L30 to P12L5:*** I do not see the value of this regime classification compared to what you already said before in this section. This could be removed from the manuscript.

**Response: The regime classification provides an overall perspective on possible conditions but, in the interest of a succinct article, this paragraph has been removed.**

***P12L31-32:*** I would be careful in interpreting these results. The precipitation patterns that you see can be caused by shifts in the most extreme storms, which cause result in areas with large increases and decreases next to each other. Shifts in the location of the precipitation maxima are typically chaotic and ensemble simulation would be necessary to identify the real climate change component of precipitation changes.

**Response: All WRF-HRCONUS information has been removed.**

***P13L10-13:*** This interpretation of the results is way too overconfident. First, how sure are you that your observed maxima is at exactly this location since you probably have a fairly sparse station network. Second, the effective resolution of climate models is typically >4-8 times its horizontal grid spacing. I would just state the model can capture the location of the precipitation maxima.

**Response: All WRF-HRCONUS information has been removed**

***P14L12:*** How can I see that in Fig. 8a?

**Response: It is correct that the information is not readily apparent in Figure 8a. To arrive at such inferences, we also utilized time series of near-surface air temperature data averaged over the eastern Prairies to compare the historical and projected end-of-the-century conditions (see figures below). For example, the projected February conditions are similar to historical conditions for March (the month during which melt typically commences historically in the area). The text has been revised to clarify the discussion.**

[Figure]

Timeseries of near-surface temperature (Tas) variables averaged over the southeastern Prairies and over different time periods: (a) Feb max Tas, (b) Mar max Tas, (c) Feb Tas, (d) Mar Tas, (e) Feb-Mar Tas and (f) Mar-Apr Tas.

**New and/or Revised Text: "Comparisons of historical and projected future surface temperatures over the eastern Prairies (not shown) suggest that spring melt would commence in February and be completed in March towards the end of the century."**

*P14L19:* what does 4.6/decade mean? Seasons per decade?

**Response: Yes, seasons/decade. The additional word has been added to the text in the appropriate locations.**

**New and/or Revised Text: "...projected to be 4.6 seasons/decade, which is substantially higher than the corresponding mean frequency of 1.2 seasons/decade for wet AMJ during 1986-2005…"**

*P14L20-21:* Why are you comparing your results to the period 2081-2100? A 2.5-degree cooler FM period at the end of the century under RCP8.5 will be much warmer than a 2.5-degree cooler FM period in the past climate.

**Response: We are actually comparing projected FM conditions to historical MA conditions. Such comparisons are based on the observation that the projected FM conditions near the end of the century are similar to historical MA conditions (see the time series plots given in the above). The text has been revised to clarify this.**

**New and/or Revised Text: "As such, the frequency of wet MAM and cool FM during 2081-2100 is compared to historical (1986-2005) wet AMJ combined with cool MA, using the anomalous conditions for the 2014 flood as criteria for each model."**

*P14L27:* Your summer cannot be typically linked to severe conditions. Severe conditions are per definition rare events and cannot be typical.

**Response: Agreed. The word 'typical' has been removed and the sentence has been altered.**

**New and/or Revised Text: "Summer across the region can be associated with severe conditions…"**

*P16L7-9:* This sounds like you are searching for an excuse to show mid-century results here. If it is that important to show mid- and end-century results you should also do this in the previous sections?

**Response: We edited the opening sentence of this Section (4.3.2) to explain why we focus on mid-century analysis for convective precipitation. Up to the time we undertook this analysis, there were no dynamically downscaled RCM data available for the end of the century covering North America and, to analyze convective precipitation, higher resolution information is needed (such as NARCCAP).**

**New and/or Revised Text: We have edited the first sentence of Section 4.3.2 to: "Analysis of future convection related precipitation requires higher spatial resolution than available from global climate models. Suitable datasets are available with dynamically downscaled RCMs, such as NARCCAP (Mearns et al., 2012). However, future scenarios are only available to mid-century when, as discussed in Sect. 3.4, dry conditions are not expected to be so dominant over the southern Prairies (Fig. 12)."**

*P16L19-30:* Why are you discussing changes in mean summer precipitation again here? Some of the results look very different to the once you show in Fig. 10d. This makes it difficult for the reader to know, which interpretation they should belief in. I would suggest to remove this paragraph and maybe add some information to section 3.4.

**Response: We have removed the total precipitation analysis (and the original Fig. 13) from this section, as the reviewer is correct. Mearns et al. (2013) have already shown NARCCAP ensemble mean summer precipitation and we have now simply compared our convective precipitation results to Mearns et al. (2013) and Mailhot et al. (2011) where possible.**

*P17L30-34:* could be removed.

**Response: We have deleted the closing paragraph text in section 4.3.2 as suggested.**

*P18L1:* This entire section has very little relevance to the CCRN region. Mentioning global average results on lightning can be easily misinterpreted as being relevant for Canada, which

they most like are not. This section can be easily condensed to a single paragraph with the main message that we simply do not know a lot about future lightning in Canada.

**Response: As suggested, the lightning section is now just one paragraph.**

**New and/or Revised: "As indicated in Sect. 3.4, convection may be enhanced or suppressed by the latter part of the century. A related issue is lightning. Since long term observations by satellite-based or ground-based lightning location systems of lightning do not exist, studies assessing past trends around the world have used thunderstorm day records (Changnon and Changnon, 2001; Pinto et al., 2013; Huryn et al., 2016) although none of these was carried out over the CCRN region. In terms of future occurrence, climate model simulations using parameterizations or proxy data for global lightning have been carried out (Price and Rind, 1994a; Romps et al., 2014; Finney et al., 2018). Although these model simulations have not been evaluated over the CCRN region, Finney et al. (2018) and Price and Rind (1994b) both projected an increase at latitudes above approximately 60°N, whereas Finney et al. (2018) projected a decrease (not statistically significant) and Price and Rind (1994b) projected an increase over parts of the Prairies. Overall, uncertainty in convection certainly translates into substantial uncertainty in lightning occurrence."**

*P19L18:* How confident are you here based on the large uncertainties in future lightning projections?

**Response: As mentioned in the original text, there is substantial uncertainty regarding future convection. We should have more clearly pointed out that the same holds for lightning.**

**New and/or Revised Text: "Overall, the uncertainty in convection certainly translates into substantial uncertainty in lightning occurrence."**

*P19L20:* What about the future availability of fuel?

**Response: No specific analysis was done on future fuels although a new reference (Flannigan et al., 2015) has been added which refers to this topic.**

**New and/or Revised Text: "Although not discussed in this article, fuel amount, type, and moisture content are important elements for fire occurrence and spread and are dependent on climate conditions Consequently, the projected summer conditions may also result in drier fuels which would also increase wildfire activity (Flannigan et al., 2015)."**

*P20L8:* You could mention Fig. 15a here.

**Response: Good point. Done.**

**New and/or Revised Text: "Figure 15a shows that, in autumn…"**

*P20L12:* Fig. 15b

**Response: Good point. Done.**

**New and/or Revised Text: "Figure 15b shows that, in spring…"**

*P20L20:* I think "link" is too strong here. You did not show any causal relationships in your analysis and your interpretation are mostly based on your interpretation of the results.

**Response:  The text has been changed to address this issue.**

**New and/or Revised Text: "The sentence now reads as follows. "We suggest that this regional acceleration is associated with the corresponding temporal behavior of the upper air large-scale drivers."**

*P20L23:* Understanding the origin of changes would be very important as well.

**Response: Absolutely. We did not specifically point out this obvious point and it has been added.**

**New and/or Revised Text: It now reads. "Additional and more comprehensive investigations on the origin and evolution of changes are certainly required."**

*P21L3:* "…natural and anthropogenic factors."

**Response: Good point. The wording has been changed.**

**New and/or Revised Text: "The atmosphere and associated features have changed and will continue to do so due to natural and anthropogenic factors."**

*P21L11:* please replace "dramatic" with something more quantifiable.

**Response: The word "dramatic" has just been deleted.**

**New and/or Revised Text: "Because of projected seasonal shifts in circulations and temperature..."**

*P21L11-12:* Which 4 models are you talking about?

**Response: We meant the four conceptual depictions in our final diagram.**

**New and/or Revised Text: "...four conceptual depictions were developed to account for changes in associated phenomena"**

Figures:

*Fig. 2,4,5,7:* The rainbow color table does not allow to see a lot of details (e.g., Fig. 2 d shows a red area along the west coast but changes in the focus region are shown in a single blue tone.

Please select a more appropriate color table and use fixed color bar ranges in all 4 figures. At the moment it is very hard to compare them. Also, adding a significant layer on top of it (similar to what the IPCC uses in their assessment) would be highly beneficial.

**Response: Thanks for the suggestion. We have improved the colour table in these plots and made the colour bars the same in each.**

**We agree that adding a significant layer on top would be beneficial. Unfortunately, due to unforeseen circumstances, we are no longer able to access the data to carry out such additional analyses.**

**New and/or Revised Text: An example of a new sub-plot:**

[Figure]

*Fig.10:* Please adjust the color range. E.g., Fig. 10a has only one red spot in the lower left corner. Gradient would be much easier to see if you would reduce the maximum from 48 to 24 hours.

**Response: This suggestion has been followed.**

**New and/or Revised Text: The new figure is shown below.**

[Figure]

***Fig. 13-14:*** Please reduce your color range also in these figures.

**Response: We have removed the original Fig. 13 but we are unsure why the existing colour range for convective precipitation is unsatisfactory. We have not changed the colour range in the revised manuscript but do mention the range in increased precipitation resulting from summer convection from the three model pairs over the Canadian Prairies in the text.**

**New and/or Revised Text: The text has been changed to:**

**"All three model pairs show increases over much of the Prairies but with varying amounts (near zero to 50 mm). MM5-CCSM and HRM3-HadCM3 are consistent with CMIP5 RCP8.5 results for the same future period (not shown) and the spatial patterns in other studies (that is, increases in Canada but decreases in central/southern U.S. Plains) (e.g. Mearns et al., 2013; Mailhot et al., 2011)."**

***Fig. 15:*** This figure could be better imbedded in your manuscript (you only mention it once).

**Response: Your suggestions were followed to more often refer to this figure.**

**New and/or Revised Text: Revised text was already shown in response to two minor points shown earlier.**

***Literature:***

Deser, C., Phillips, A., Bourdette, V. and Teng, H., 2012. Uncertainty in climate change projections: the role of internal variability. Climate dynamics, 38(3-4), pp.527-546.

Trenberth, K.E., Dai, A., Rasmussen, R.M. and Parsons, D.B., 2003. The changing character of precipitation. Bulletin of the American Meteorological Society, 84(9), pp.1205-1218.

Prein, A.F., Rasmussen, R.M., Ikeda, K., Liu, C., Clark, M.P. and Holland, G.J., 2017. The future
intensification of hourly precipitation extremes. Nature Climate Change, 7(1), p.48.

Musselman, K.N., Lehner, F., Ikeda, K., Clark, M.P., Prein, A.F., Liu, C., Barlage, M. and Rasmussen, R., 2018. Projected increases and shifts in rain-on-snow flood risk over western North America. Nature Climate Change, 8(9), p.808.

Dai, A., Rasmussen, R.M., Liu, C., Ikeda, K. and Prein, A.F., 2017. A new mechanism for warm season precipitation response to global warming based on convection-permitting simulations. Climate Dynamics, pp.1-26.

Mearns, L.O., Sain, S., Leung, L.R., Bukovsky, M.S., McGinnis, S., Biner, S., Caya, D., Arritt, R.W., Gutowski, W., Takle, E. and Snyder, M., 2013. Climate change projections of the North American regional climate change assessment program (NARCCAP). Climatic Change, 120(4), pp.965-975.

Gutowski Jr, W.J., Arritt, R.W., Kawazoe, S., Flory, D.M., Takle, E.S., Biner, S., Caya, D., Jones,
R.G., Laprise, R., Leung, L.R. and Mearns, L.O., 2010. Regional extreme monthly precipitation simulated by NARCCAP RCMs. Journal of Hydrometeorology, 11(6), pp.1373-1379.

**REVIEWER 2**

This review is written by researchers with ample experience in the Canadian climate. The manuscript aims at synthesizing expected changes in future regional climate of western Canada due to anthropogenic influences. The authors provide a wealth of information selecting specific processes or phenomena of relevance to the region. In doing so, they define the basis for research priorities to advance on the knowledge of the regional impacts of a changing climate. I found the review and synthesis to be useful, exhaustive and well documented.

As with many review articles written by several authors, there tends to be some inconsistency among the different sections, with some easier to read than others. For example, the motivation section is clear, defined and even entertaining. On the other hand, section 3 could benefit from refining the main concepts and doing a better link between the discussion and the supporting figures. The manuscript has value and quality. It will be even more attractive once the sections that need improvement are refined.

I recommend that the manuscript is approved after the following points are addressed.

1. Section 3 seems to lack an introduction and jumps directly to describe changes in large-scale seasonal patterns. As the discussion is based on changes or anomalies, it would be useful to start with a description of the present climate and more specifically of the PNA pattern as known now. This section also assumes that many features are known to most readers. The second paragraph in page 5 is an example of frequent statements presented without a clear argument: "Projected regional climate responses to the circulation changes are consistent with those found during negative PNA, but shifted in association with the projected circulation features." Not everybody knows the regional features of the negative PNA, or how they would be shifted due to changes in the projections. Is there a way to infer the changes in cold advection from the figures? Unfortunately, this is not a matter of rewriting a couple of paragraphs. Rather, it is about how the (quite complex) concepts of the PNA pattern and its future changes are presented for the four seasons. My suggestion is that the authors simplify the text by limiting the discussion to key issues that can be easily linked to the figures or adequate references. I suggest following a similar approach to that in section 4.1.1. The discussion of changes in the 0 C isotherm is straightforward and supported by a figure that is easy to follow (Fig. 8).

**Response: We appreciate these comments. We now include a brief summary of large scale conditions within the present climate and large scales including PNA. Cold air advection is inferred from the presence of pressure anomalies driving flows from the north. Improvements in the text have been made to improve clarity in the discussion.**

**New and/or Revised Text: Text has been updated or added in a few locations as follows:**

**Brief summary of circulations and associated regional impacts in current climate: Section 3 new introductory paragraphs: "As we are focusing the discussion on projected changes of the PNA pattern and quasi-stationary upper air circulation features over the northwestern U.S., it is appropriate to briefly summarize how these large-scale drivers affect the current**

**hydroclimate climate over western Canada during the cold- and warm seasons, respectively.**

**The autumn and winter positive (negative) PNA pattern is characterized by large-scale upper-level negative (positive) and positive (negative) height anomalies centered above the Aleutian Islands and the western Canadian Prairies, respectively. At the surface, a broad anomalous low (high) centered just south of the Aleutians and extending into the Mackenzie basin is typically found during the positive (negative) phase. The warm (cold) temperature advections associated with the low-level southwesterly (northwesterly) anomalous flow typically induces warm (cold) temperature anomalies over northwest Canada during positive (negative) PNA. In addition, dry (wet) conditions over the western Prairies are associated with the positive (negative) pattern.**

**As shown in Shabbar et al. (2011), Brimelow et al. (2015) and Szeto et al. (2015, 2016), the tracking and development of synoptic systems that significantly affect the warm-season hydroclimate of southwestern Canada are strongly affected by the large-scale upper-level pressure anomaly over the northwestern U.S. In particular, anomalously wet (dry) conditions are typically found to be associated with upper low (high) pressure anomalies over the region."**

**Section 3.1: "Similar circulation patterns are typically found during negative PNA conditions with an anomalous low centered above the southern Prairies, and a high above the Aleutians."**

**Section 3.2: "Some changes at the lower levels (Fig. 4b) are similar to those found in positive PNA conditions with an anomalous trough extending from the Aleutians into areas off the west coast of North America. But, a strong anomalous surface ridge that is typically located over the western U.S. under positive PNA conditions is projected to be centered over southwestern Canada."**

2. I have difficulty agreeing with the interpretation of Figure 13. "Consistent" features are described for relatively small regions of the domain (e.g., central and northern Manitoba to north central Alberta), but the main issue that seems to be ignored is that there are important inconsistencies over large areas of the domain. These large-scale differences among the models can suggest that agreements over the small regions are just the result of chance. An objective approach is needed to separate the wheat from the chaff.

**Response: We have removed the total precipitation analysis from this section 4.3.2. as it is somewhat redundant to Mearns et al. (2013). We have edited this section to only discuss convective precipitation and hail. Below is part of our edits that is relevant to the reviewer's comment.**

**New and/or Revised Text: "NARCCAP historic (1971 to 2000) and mid-century future (2041 to 2070) model output was used to assess future changes in convective precipitation and hail over the Canadian Prairies, southern Northwest Territories and U.S. northern plains. Convective precipitation is defined to occur when the model convective scheme is**

triggered to release latent energy and convective instability through simulated vertical motion. Brimelow et al. (2017) suggested that the three most consistent NARCCAP model pairings to assess convective precipitation and hail for the regions of interest herein, included MM5-HadCM3, MM5-CCSM and HRM3-HadCM3, based on their ability to reproduce the precipitation climatology. No other NARCCAP studies focused on hail or warm season convection-only precipitation, although Mearns et al., (2013) and Mailhot et al., (2011) looked at ensemble summer total precipitation and annual maximum precipitation, respectively, while other studies focused on other seasons (e.g. Gutowski et al., 2010; Kawazoe and Gutowski, 2013).

Changes in future summer (JJA) convective precipitation are shown in Fig. 13 for three NARCCAP model pairs under the SRES A2 scenario. All three model pairs show increases over much of the Prairies but with varying amounts (near zero to 50 mm). MM5-CCSM and HRM3-HadCM3 are consistent with CMIP5 RCP8.5 results for the same future period (not shown) and the spatial patterns in other studies (that is, increases in Canada but decreases in central/southern U.S. Plains) (e.g. Mearns et al., 2013; Mailhot et al., 2011). These results are also consistent with general increases in CAPE and surface dew points in a warming climate over much of the Prairies (e.g. Brimelow et al., 2017)."

3. It is discussed that given the lack of lightning data, a proxy based on cloud-top heights has been used by Price and Rind (1994). Has this approach been validated in any manner?

**Response: We now specifically mention that no evaluation has been carried out over the CCRN region.**

**New and/or Revised Text: "Although these model simulations have not been evaluated over the CCRN region, Finney et al. (2018) and Price and Rind (1994b) both projected an increase at latitudes above approximately 60°N, whereas Finney et al. (2018) projected a decrease (not statistically significant) and Price and Rind (1994b) projected an increase over parts of the Prairies."**